# The Large and Diverse Family of Mediterranean Flat Breads: A Database

**DOI:** 10.3390/foods11152326

**Published:** 2022-08-04

**Authors:** Antonella Pasqualone, Francesca Vurro, Carmine Summo, Mokhtar H. Abd-El-Khalek, Haneen H. Al-Dmoor, Tomislava Grgic, Maria Ruiz, Christopher Magro, Christodoulos Deligeorgakis, Cynthia Helou, Patricia Le-Bail

**Affiliations:** 1Department of Soil, Plant and Food Science, University of Bari Aldo Moro, Via Amendola, 165/a, 70124 Bari, Italy; 2Brussels Institute of Advanced Studies (BrIAS) Fellow 2021/22, Elsene, 1050 Brussels, Belgium; 3Food Technology Research Institute (FTRI), Agricultural Research Center (ARC), Giza 12619, Egypt; 4Scientific Food Center (FACTS), Princess Taghreed Street Bulding 68, P.O. Box 177, Amman 11831, Jordan; 5Faculty of Food Technology and Biotechnology, University of Zagreb, Pierottijeva 6, 10000 Zagreb, Croatia; 6Institute of Agrochemistry and Food Technology (IATA-CSIC), C/Agustin Escardino, 7, 46980 Paterna, Spain; 7Department of Food Sciences and Nutrition, Faculty of Health Sciences, University of Malta, MSD2040 Msida, Malta; 8Department of Food Science and Technology, Alexandrian Campus, International Hellenic University (IHU), 57400 Thessaloniki, Greece; 9Department of Nutrition, Faculty of Pharmacy, Saint Joseph University of Beirut, Beirut 1004 2020, Lebanon; 10INRAe, UR1268 Biopolymères, Interactions, Assemblages (BIA), Rue de la Géraudière, CEDEX 3, 44316 Nantes, France

**Keywords:** flat bread, flour quality, traditional bread, ethnic food, baking system, vertical oven, bread culture, food heritage, food diversity, quality schemes

## Abstract

An in-depth survey was conducted by collecting information from web sources, supplemented by interviews with experts and/or bakers, to identify all the flat breads (FBs) produced in the nine Mediterranean countries involved in the FlatBreadMine Project (Croatia, Egypt, France, Greece, Italy, Jordan, Lebanon, Malta and Spain), and to have an insight into their technical and cultural features. A database with information on 143 FB types (51 single-layered, 15 double-layered, 66 garnished, 11 fried) was established. Flours were from soft wheat (67.4%), durum wheat (13.7%), corn (8.6%), rye, sorghum, chickpea, and chestnut (together 5.2%). The raising agents were compressed yeast (55.8%), sourdough (16.7%), baking powder (9.0%), but 18.6% of FBs were unleavened. Sixteen old-style baking systems were recorded, classified into baking plates and vertical ovens (*tannur* and *tabun*). Artisanal FBs accounted for 82%, while the industrial ones for 7%. Quality schemes (national, European or global) applied to 91 FBs. Fifteen FBs were rare, prepared only for family consumption: changes in lifestyle and increasing urbanization may cause their disappearance. Actions are needed to prevent the reduction of biodiversity related to FBs. Information in the database will be useful for the selection of FBs suitable to promotional activities and technical or nutritional improvement.

## 1. Introduction

Bread is one of the basic components of the daily diet, and its history is linked to human history. The “flat” breads include a multitude of bread types different from each other but always relatively thin, ranging from a few millimeters to a few centimeters in thickness. These breads, whose origin is very ancient [1], are produced all over the world. Those spread from the Fertile Crescent reached the Mediterranean area (North Africa, Southern Europe, Middle East and Anatolian peninsula), the Indian subcontinent, and the Caucasian region, up to Xinjiang [2], as well as the Arabian peninsula, with interlinks with the Horn of Africa [2,3]. Flat breads are also produced in the American continent, mostly in Central and South America [4], but also in the North, owed to Native Americans [5].

Flat breads meet the need of increasing the sustainability of food system for several reasons: (i) Can be obtained from cereals other than wheat, as well as pseudocereals or pulses, allowing the use of local productions from marginal lands; (ii) require short baking times, eventually even without using an oven (under hot ashes); (iii) can wrap around food or serve as a spoon, reducing tableware use and water consumption; (iv) are transported with little encumbrance and reduced energy impact; (v) if baked to dryness, have a quite long shelf life, reducing bread waste [2].

These strong points made flat breads very popular so that, though having an ancient origin, they have survived until today. Nowadays, these highly versatile breads can be produced either in the same way as they were made thousands of years ago or in modern fully automatic industrial lines. In addition, they can be seasoned or stuffed with a variety of ingredients becoming cheap, convenient, and palatable street foods. Renowned examples of these are the *döner kebab* and the related *shawarma* and *gyro*, i.e., finely sliced roasted meat rolled, or stuffed, in a pocket-type flatbread (typically named “*pita*”) with salad and various sauces [6,7]. Other examples are the Italian *pizza* and *focaccia*, or the French *fougasse*, prepared by seasoning the surface of the flattened dough disc with several ingredients, before baking [8]. The fast pace of the modern lifestyle has led to a growing demand for ready-to-eat foods and a concomitant increase in the consumption of flat breads. The global market for these products accounted for $81,796.6 million in 2018 and is expected to grow to $145,180.9 million by 2027 [9].

Although flat breads are an ancient and consolidated product, there is still much room for technical and nutritional improvement. The baking step is typically very fast, being carried out at high temperature with direct heating. This process may cause quality and safety issues such as burned edges, due to non-uniform heat distribution, and the formation of combustion contaminants such as benzopyrenes and polycyclic aromatic hydrocarbons (PAHs) [7,10]. New baking systems have been recently proposed, such as an indirect heating plant with a rotary baking tray [11]. Moreover, the Bake Off Technology (BOT), consisting of producing bread from industrial refrigerated, frozen or non-frozen bakery goods (partially-baked bread or “part-baked” bread) to be sold for domestic baking, has increased its market share indicating a growing interest by consumers [12], and could be applied to flat breads. Flat breads, which are a staple food of high nutritional importance in many countries, are also suitable for reformulation with a variety of fortifying ingredients of animal or plant origin, able to raise the content of proteins and micronutrients [13,14].

In this context, an international research project, namely “Flat Bread of Mediterranean area: Innovation & Emerging process & technology” (FlatBreadMine) has been recently financed by the European Union H2020-PRIMA, with the main aim of valorizing and innovating flat breads. However, to propose technical innovations (such as low-pressure baking and part-baking) and nutritional improvement (by incorporating flour of legumes, acorns, or carobs), a precise picture of the existing flat breads is needed, in order to select the most suitable ones.

The aim of this work was, therefore, giving an insight into the technical and cultural features of the flat bread types produced in each one of the Mediterranean countries involved in the FlatBreadMine project (namely Croatia, Egypt, France, Greece, Italy, Jordan, Lebanon, Malta and Spain). The steps to achieve this goal are: (1) To identify all the flat bread types produced in the selected countries; (2) to collect information on their main technical characteristics, from the starting ingredients to the end-product, including the cultural features; (3) to set up a database containing all the information; (4) to examine and interpret the information collected [15] in order to highlight the diversity of flat breads across the considered countries and to define the selection criteria.

## 2. Materials and Methods

### 2.1. Surveyed Area

Nine countries of the Mediterranean area, involved in the FlatBreadMine project, were objects of study: Croatia, Egypt, France, Greece, Italy, Jordan, Lebanon, Malta, and Spain.

### 2.2. Subject of the Survey

A survey was carried out to collect information on flat breads, including traditional and artisanal ones. The surveyed flat breads had to be original and native of each surveyed country. Therefore *pizza*, for example, which has an Italian origin, was listed as an Italian flat bread and surveyed only in Italy, although prepared also in the other countries object of the study.

For each flat bread, the following data were collected: (i) The regional area or town of origin, and the area of marketing and diffusion; (ii) the ingredients used in bread preparation (flour, yeast, additional ingredients, and their ratio); (iii) the raw material characteristics (type of flour and its quality features; type of yeast; information on any additional ingredient); (iv) the production process, step by step (kneading time and temperature; conditions of the first leavening step; shaping specifications in terms of average diameter and thickness; conditions of the second leavening step; time and temperature of baking; oven type); (v) the characteristics of bread (type and size; optimal quality features; artisanal or 35-industrial); (vi) the main references and sources of information.

### 2.3. Data Collection

Data were collected between October 2021 and May 2022. The first step was the identification of all flat breads produced in each country, which was done by accessing the websites of bakers’ associations, food blogs, and scientific literature; browsing the official lists of traditional food products uploaded onto the websites of the EU, the Italian Ministry of Agriculture, and Slow Food; consulting local experts. The latter were scholars involved in the protection and rediscovery of traditional food products including bread, who helped to uncover rare breads not regularly available in the market. They were selected via convenience sampling, based on direct knowledge with the researchers involved in the study, and were contacted by phone for advice.

The second step consisted in collecting the information referred to in Section 2.2. for each flat bread. Information was primarily retrieved from web sources: official technical datasheets of breads awarded of quality marks, scientific literature, websites of bakers’ associations, and food blogs. Missing information in web sources was sought from the experts and/or from bakery managers/owners through structured face-to-face/phone interviews. The recruitment of respondents (experts and/or bakers), according to convenience sampling, was based on direct knowledge with the researchers involved in the study, or supported by the bakers’ associations, who introduced the researchers. Interviews were based on a questionnaire (Appendix A) composed by qualitative and quantitative open-ended questions, which was pre-tested with the president of the consortium of bakers specialized in the production of *Focaccia barese* flat bread (Bari, Italy), who was asked to answer the questions and comment on their feasibility, to avoid excessively generic, or too specific and technical questions, difficult to understand. After pre-testing, technical questions regarding bread packaging, modified atmosphere and storage conditions were deleted. To reduce work, considering the great number of surveyed breads (*n* = 143), only the questions necessary to fill the information gaps with respect to web sources were asked. In addition, experts who knew more than one type of flat bread, as well as bakers who produced more than one type of flat bread, were asked to provide information on all of them. The first contact was by telephone, to present the study and to make an appointment for the face-to-face or phone interview, if the participant agreed. The choice between face-to-face and phone interview was made according to the interviewee’s preference. The interviewers were the researchers involved in the study. They facilitated the comprehension, also linguistic, of the questionnaire, which was written in English. The interviewers let the conversation flow naturally and took notes of the answers to each question. Data were anonymized and treated in an aggregated way.

### 2.4. Database Structure

A database was set up by the Excel software (Microsoft Office, Version 2018 for Windows, Microsoft Corporation, Redmond, Washington, DC, USA) to gather information on the flat breads of each surveyed country. The database structure included one row per each bread type and 27 columns for the above reported information. In addition, a representative picture was included for each type of bread.

The database was uploaded onto the FlatBreadMine project website and is publicly accessible at the link: https://flatbreadmine.eu/resources/ (accessed on 7 July 2022).

### 2.5. Data Analysis

The distribution of data was analyzed as percent frequency by Excel software (Microsoft Office, Version 2018 for Windows, Microsoft Corporation, Redmond, Washington, DC, USA).

## 3. Results and Discussion

### 3.1. Flat Bread Diversity in the Surveyed Area

A total of 143 different flat bread types were found to be produced in the surveyed countries, distributed as follows: 14 from Croatia, 8 from Egypt, 3 from France, 23 from Greece, 75 from Italy, 6 from Jordan, 7 from Lebanon, 2 from Malta, and 5 from Spain (Figure 1) [16].

The high number of flat breads recorded in Italy was probably due to the existence of strong regional gastronomic differences within the Italian territory. Furthermore, the consolidated tendency to rediscover and keep alive the memory of small-scale, local, and traditional food products, included flat breads, finds its maximum expression in Italy, home of the Slow Food Foundation for Biodiversity [17]. This trend, appreciated by modern consumers [18,19], aligns with the European policies for promoting traditional foods and protecting their origin [20], and will be discussed more in depth in Section 3.9.

Another important factor is the different meaning that flat breads assume in different areas. In Italy these products are considered as a delicacy, admitting many variations on a regional basis, while in the areas where flat breads originated in the antiquity, i.e., Middle East [1,2] (Jordan and Lebanon, for this survey), or in closer countries, such as Egypt, they represent staple foods that are consumed daily, therefore are less affected by variations.

Flat breads can be classified into plain (further categorized into single- or double-layered), garnished (seasoned or stuffed), and fried. Table 1 shows the occurrence of flat breads in the different categories through the surveyed countries. Garnished flat breads accounted for 46.2%, with a 27.3% contribution by Italy. These flat breads, prepared as specialties to be consumed occasionally, were seasoned or stuffed with several ingredients before baking. In Jordan and Egypt, instead, flat breads were only plain.

Among the plain types, the single-layered category, easier to be prepared, accounted for 35.7%. The double-layered (10.5%) are, instead, those characterized by the typical “pocket”, such as the Jordan *Kmaj* [21] (Figure 2A), the Egyptian *Shamy* and *Baladi* [22], and the common Arabic bread or *Khobz* (*Khobz* means “bread” in Arabic) (Figure 2B), which are all known in western countries as “*Pita*” bread.

The pocket is the visible result of the thermal expansion of the fermentation gas into a thin dough layer, which takes place during baking (Figure 3).

Fried flat breads accounted for 7.7%. They were recorded in Italy (6 breads, namely *Gnocco fritto*, *Crescenta fritta*—also named *Crescentina fritta*—*Pinzini ferraresi*, *Cresciolina*, *Pizza fritta*, *Pitt’ajima*), Greece (4 breads: *Pisia*, *Tiganopsomo*, *Sfakianopita* and *Fyllota*), and Spain (one bread: *Arepas Canarias*).

Figure 4 shows, per each country, the number of flat bread types marketed outside the area of origin (town, subregion), compared to the total number flat breads produced in the country. In Italy, Greece, and Croatia, only a minority of flat breads were found to be marketed through the entire country, outside the area where they are originally produced and consumed, which was generally a very limited geographic area.

On the contrary, in countries such as Jordan, Lebanon, and Egypt, the distribution and consumption of flat breads was homogenous throughout the entire country and three flat breads from these countries were also exported abroad (namely the Egyptian *Baladi* and *Shamy*, and the Lebanese *Khobz*). These findings agreed with the different character, previously discussed, that flat breads assume in different areas: local specialties vs. national staple foods.

### 3.2. Flour Type and Quality

Refined soft wheat flour was widely used (67.4%) in the preparation of the surveyed flat breads (Table 2).

Whole meal flour of soft wheat was used only in 5.1% of cases. Earlier studies, dating the late nineties, reported that flat breads were commonly made of wheat flour at high extraction levels [4], so this flour has been progressively substituted by the refined one.

The use of durum wheat flour (more precisely, re-milled semolina) was reported in 13.7% of cases. Durum wheat cultivation is common in semiarid zones of the Mediterranean basin, and its use in bread making has been already reported [23]. In fact, durum wheat breads (not flat), such as Altamura bread [24] and Dittaino bread [25], have been awarded by the Protected Designation of Origin (PDO) European Union (EU) mark for their peculiar characteristics, such as a compact and yellowish crumb (due to carotenoid pigments). Durum wheat whole meal represented an exception and was found only in the preparation of the Greek *Koulouri*.

The use of corn refined flour accounted for 8.6%. In three cases it was subjected to thermal treatments: pre-cooked corn flour was recorded in the production of the Spanish *Arepa Canarias*, and up to 30% extruded-cooked corn flour could be optionally added to soft wheat refined flour to prepare the Croatian flat breads *Pogača* and *Kukuruzna miješana ciabatta* (“corn composite *ciabatta*”). The thermal treatment causes starch gelatinization, increasing dough viscosity [26]. In the presence of wheat flour, this effect is not strictly needed because good viscoelasticity is ensured by gluten; however, the thermal treatment could be made also because it is able to slow down bread aging [27,28].

Corn has long been cultivated in the Canary islands and Eastern Europe, including Croatia [29], so in the past the exclusive use of corn in the preparation of these breads could be hypothesized, explaining the need of pre-gelatinizing flour. The use of pre-cooked corn flour in the preparation of *Arepa Canarias*, indeed, resembles the procedure adopted for its Venezuelan counterpart, *Arepa*, which is made from corn only [30]. Probably a return cultural contamination took place in the Canary islands following migrations to America, including Venezuela.

Without a thermal pretreatment, instead, corn flour is used in the preparation of the Egyptian *Bataw* and *Meraharah*, as well as in the *Talo*, a traditional bread from the Mungia subregion of Basque Countries. Interestingly, the latter has been associated by archeobotanists to an ancient flat bread of the same area, made of acorn [31], which is being rediscovered recently for its high nutritional value [32]. Corn flour can be used as an alternative to wheat flour in the Greek *Souvlakopita* and *Plakopita*, while *Bobota*, which was the most consumed bread in Greece during the German-Italian occupation during World War II, was and is still made exclusively from corn flour [16]. In Italy, corn flour is used to prepare the *Carchiola*, *Torta al Testo*, *Pizza di granone*, *Pizza con farina di mais*, and *Pizza di farinella bacolese*. Only a very small amount of corn flour, about 5%, is mixed with wheat flour for preparing the Lebanese *Markouk*.

Rye flour was found to be used in colder areas—to which is more adapted [33]—such as the alpine Italian regions and part of Croatia. A mixture of rye and soft wheat flour, indeed, is used to prepare the Italian *Puccia ladina* (typically consumed in the mountain huts of the Dolomites) and *Schüttelbrot*. Rye flour is optionally added to soft wheat flour to prepare the Croatian *Pogača* and *Kruh ispod peke* (“bread under the lid”).

Like other commodities, cereals and particularly wheat, have long since become fully globalized. Egypt, for example, has become the world’s largest importer of wheat, exposing the country to significant vulnerabilities, not to mention that more than half of the consumption of wheat in the Mediterranean countries comes from Russia and Ukraine [34]. The use of alternative flours should therefore be strengthened. Among them, sorghum flour was traditionally used to prepare the Egyptian *Khobz min el dorra al rafi’ah* and *Zallut*; however, these breads are now only prepared at home for family consumption and not for commercial purposes, with a serious risk of losing their knowledge.

Pulses, though nutritionally valuable, with an amino acid profile complementary to that of cereals, are underrepresented. Chickpea flour is used only in the Italian *Farinata*, a typical street food from Liguria region with variants in Tuscany and Piedmont [35]. However, the addition of pulse flours to bakery products, including flat bread, has been the object of a rising attention in the recent years [36,37,38,39].

Chestnut flour is used for preparing the Italian *Neccio*. Besides their high antioxidant activity, chestnuts are rich in minerals, polyunsaturated fatty acids, fiber, and vitamins [40]. In Italy, indeed, this crop has an important economic value [41] and specific chestnut cultivars (Carpinese, Pontecosi, Capannaccia, and Morona) grown in the Garfagnana subregion of Tuscany are used to prepare the flour named “*Farina di Neccio della Garfagnana*” which has been recognized as a PDO food product. According to tradition, before milling, the chestnuts are dried in small stone buildings named “*metati*”, where a hearth on the ground floor heats and dries the chestnuts placed on the upper floor [42].

Regarding the quality of flours used for flat breads, a limited technical knowledge was observed in all the surveyed countries, especially (but not exclusively) for the most artisanal productions. Consequently, this information often remained undefined in the database (where “not specified” is indicated), reaching 71.6% of missing data, which was the highest percentage among all collected data. Where information on the quality features was available, protein and gluten content and gluten quality were the most frequently reported. For refined wheat flour the quality parameters generally were: protein content ≥ 9%, wet gluten content ≥ 25%, alveograph W ≥ 180 × 10^−4^ J, and P/L ≤ 1. For *Pizza Napoletana* Traditional Specialty Guaranteed (STG) more detailed quality information was available: dry gluten 9.5–11%, alveograph W 220–380 × 10^−4^ J, P/L 0.50–070, water absorption 55–62%, farinograph stability 4–12 min, farinograph drop off of consistency ≤60 Brabender Units [43]. For corn flour only protein content ≥7% was reported.

It should be mentioned that data collection at bakers faced obstacles related to the COVID-19 pandemic restrictions and to the successive flour shortage following the Ukraine crisis. The economic value of bread and its scarcity have always important social repercussions, as evidenced by past and recent history. Bread availability provides a sense of security, while the lack of bread can be the cause of violent social movements. For example, in Lebanon, where the COVID-19 pandemic economic loss overlapped with a pre-existing crisis [44], further wheat shortage exacerbated the situation, making wheat and bread availability an extremely hot topic. Especially small producers suffered from the financial crisis on full blow, which took its toll on their sales to the point that they cannot afford even proper maintenance for their equipment. A similar situation was observed in Egypt and Jordan [45,46]. At various levels, lockdown-related economic losses and flour shortage were common to all countries, making bakers not really inclined to cooperate with the interviews. These issues were reduced, but not totally solved, with the help of local associations of bakers which introduced the researchers, or relying on direct knowledge with them.

### 3.3. Additional Ingredients

Though many plain flat breads (54.3%) did not contain any lipid, olive oil was used in 19 cases, 10 of which were characterized by the use of the extra virgin category (Table 3).

Two flat breads included sunflower oil in their formulation, and three were prepared with lard, namely the Italian *Piadina Romagnola*, *Crescentina di Modena,* and *Torta al testo,* whose official technical sheets [8,47,48,49], however, report also the possibility to use olive oil. The use of lard is traditional in the area of origin of these three flat breads, which is approximately the same area of *Prosciutto di Parma* PDO ham and is characterized by the presence of numerous pig farms.

However, besides the obvious nutritional and health implications related to the reduction of saturated fatty acids, the substitution of lard with olive oil (possibly extra virgin olive oil), could eventually overcome religious restrictions for pork-derived ingredients.

All the garnished flat breads contained vegetable oils or lard. Vegetable oils, when specified, were represented by olive, sunflower, or rapeseed oil, and their blends (Table 4). Greek, Italian, and French garnished flat breads were prepared with olive oil, and in particular extra virgin olive oil was used in the Italian *Focaccia di Recco* and *Pizza Napoletana* STG, agreeing with their official technical sheet [43,50]. Sunflower oil was used in the Croatian *Rudarska greblica* and *Zlevanka*, as well as in the *Poljički soparnik*, where, however, it was used in 50:50 mixture with olive oil. The French *Flammekueche*, of Alsatian origin (with German influence), was prepared with rapeseed oil. Lard, instead, was used in two Croatian (*Rudarska greblica* and *Zlevanka*) and ten Italian garnished flat breads (*Gnocco ingrassato*, *Focaccia Novese*, *Focaccia di Voltri*, *Crescia d’la stacciola*, *Crescia brusca*, *Pizza a scannatur di Carbone*, *Scarcedda*, *Pizza con i cingoli di maiale*, *Pizza con le sfrigole*, *Crescenta bolognese*).

During kneading and baking several reactions take place, including lipid oxidation. Studies carried out in several types of Italian *focaccia* have shown that the level of oxidation may change by varying the type of toppings, with moist ingredients able to mitigate the rise of temperature during baking, thus exposing the lipid fraction to a less severe heat stress [8].

Excluding oils and fats, the garnished flat breads contain also other ingredients which impart them well-defined and recognizable sensory characteristics and, by varying in nature and quantity, differentiate them into a myriad of nuanced variations. These ingredients are used to stuff or season the dough before baking, and their combination generates a pleasant palatability which makes the final product become much more than a staple food. The additional ingredients can be of plant and/or animal origin. The plant-based ones, which include spices, various vegetables, cereals, or seeds, accounted for 40.5%, while the sum of the ingredients having animal origin (dairy products, eggs, cured meat, canned fish, and meat) totalized 59.5%. Cured meat includes several charcuterie products such as salami and ham, while fresh meat is typically pork meat, added for example to Italian *Pizza a scannatur di Carbone* prepared on the day of the pig slaughter.

### 3.4. Leavening

Leavening agents were: compressed bakers’ yeast (*Saccharomyces cerevisiae*) (55.8%), sourdough (16.7%), and baking powder (9.0%), while 18.6% flat breads were unleavened (Table 5).

Sourdough, indeed, despite being the most traditional leavening method in the past, has overtime been partly replaced by compressed yeast, which is easier to handle and reduces leavening times. However, there is a renewed interest by consumers toward sourdough-leavened flat breads, also based on research results that prove nutritional and qualitative improvements, such as a reduction of phytic acid in whole meal flat breads [51], an increase of selenium bioavailability [52], and improved shelf life [53], texture and sensory properties [54,55].

More “modern” leavening aids, such as baking soda, are used in the Italian flat breads *Piadina Romagnola*, *Crostolo*, *Torta al Testo*, *Pizza scima*, *Gnocco fritto*, *Crescenta fritta*, in the Croatian *Pogaca z oreji* and *Zlevanka*, and in the Greek *Plakopita* (*Pita* bread baked on a stone). In the preparation of *Crescentina di Modena* baking soda can be used as an alternative to the most used compressed yeast. Similarly, the Egyptian *Shamy* bread is leavened with baking powder as an alternative to compressed yeast. In three Jordan breads (*Mashrouh*, *Tannur*, and *Taboun*), instead, baking soda is used together with compressed yeast.

Among the unleavened flat breads there is an Italian one whose multiple names all remind the absence of fermentation: “*Pizza azzima*” (which is the original name, literally meaning “unleavened pizza” in Italian), and its naming variations “*Pizza ascima*”, “*Pizza scive*” and “*Pizza scime*”. Another unleavened Italian flat bread, similar to the former, is the “*Pitt’ajima*”, whose name is also clearly related to the original “*Pizza azzima”*.

The Egyptian *Shamsi* bread (whose name derives from the Arabic word “*shams*”, meaning “sun”), instead, is a leavened one. Its leavening phase is interesting from the ethnographic point of view, because traditionally it takes place with the help of sun heat, i.e., open air, under the direct sunlight [16] (Figure 5).

This bread is decorated by making three crescent-shaped cuts, that form three angles as the dough rises. Coptic Christians, instead, make four cuts to obtain a roughly cross-shaped loaf [56] (Figure 6).

### 3.5. Baking

When specified, the declared baking temperatures were <250 °C (50.3%), between 250 and 300 °C (7%), and >300 °C (28.7%) (Table 6).

In the past, flat breads were baked only in wood-burning ovens, at very high temperatures, above 300 °C. Nowadays, the adopted baking temperature tends to be lower, below 250 °C, due to greater awareness on the risks related to the formation of thermal contaminants, such as polycyclic aromatic hydrocarbons (PAHs) [10,57] and acrylamide [58].

Besides the modern electric or gas ovens (belt conveyor tunnel ovens or batch ovens), several traditional baking systems are still used in the preparation of flat breads. This survey recorded 16 different traditional ways to bake flat bread (Table 7).

The method that most resembles the way flat breads were probably baked in antiquity was recorded for the Jordanian *Arbood*, the traditional bread of the Bedouins. This bread is prepared in the easiest possible way, i.e., unleavened and baked under hot ashes (a very rational way to cook, when an oven is not available). A fire is lit in a sandy area and, after the wood has burned, the dough disc is placed over hot ashes and covered with other ashes. During baking, bread is turned with the help of a stick, to cook evenly on both sides.

Another simple baking tool is a metal grill placed on the embers. The one used for baking the Italian *Carchiola*, named *r’ticula*, has a pivot in the center, so that it can be turned without removing it from the embers of the fireplace. The convex circular metal griddle (named *Saj* in Middle East and Egypt and *Satsi* in Greece) (Figure 7) is used in a similar way to the metal grills, being placed on the embers. It has a large diameter, approximately 50 cm, and it is suitable to bake very large flat breads such as the Jordan *Shrak* (also named *Saaj* bread from its baking system) and *Mashrouh* and the similar Lebanese *Markouk* (named also *Saj* bread).

The most traditional types of oven are those which retained the greatest differences among countries, being tightly related to local history and habits. Basically, two main baking systems were observed: baking plates (originally made of raw clay or terracotta, but nowadays generally substituted by iron cast), either coupled with lids or not, and vertical ovens: *tannūr* (transcribed also as *tannur* or *tannour*; pl. *tananir*) and *tabūn* (or *tabun*, *taboun*; pl. *tawabeen)*.

Baking plates, i.e., large, open and shallow vessels, have been documented for cooking food in various contexts—temporal, spatial, and cultural, in parts of Europe and the eastern Mediterranean [59]. Late Neolithic baking pans of the Balkans (early 5th millennium B.C.) and the subsequent baking plates of central Europe (late 5th and early 4th millennia B.C.), are small- and medium-sized flat trays interpreted as being used for baking bread [60]. Furthermore, baking trays (25–40 cm diameter) with elaborate molded rims, appear in Syria during the Early Bronze Age (3200–2000 B.C.) and are found throughout the Levant during the Middle (2000–1600 B.C.) and Late Bronze Ages (1600–1100 B.C.). They are interpreted as vessels used on special occasions for baking bread or flat pies [61].

Baking lids have a documented ancient root in the Roman *testum.* Indeed, Cato reports in the *Liber de agri cultura* (chapters 74 and 75) that bread and other foods were baked “*sub testum*”, i.e., on the hearth and under a terracotta lid named *testum* [62]. Instead, terracotta baking plates to be placed on the embers, named *testelli* (sing. *testello*), were reported in the Middle Age in central-northern Italy [62]. The influence of the original Latin word on the name “*Testo*” given to the terracotta plate with a terracotta lid traditionally used to bake the Italian *Testarolo Pontremolese* and *Torta al Testo* is evident, as is the influence on the name of the corresponding flat breads.

Another naming similarity is between *tigella* (pl. *tigelle*), the terracotta plate used to bake the *Crescentina di Modena* (which is very often named also *tigella*, after its baking system), and *taguella*, the typical flat bread prepared by the Touareg of Central Sahara [63]. The *tigella* terracotta plate derives its name from the Latin verb *tegere*, meaning “to cover”, because the traditional way of cooking the *Crescentina di Modena* involved to place the dough on the red-hot *tigella*, to cover it with another *tigella*, and to stack several of them in the fireplace. Chestnut or walnut leaves were used to avoid the direct contact of the dough with terracotta, as well as to flavor it and keep it clean from the ash.

*Tannur* and *tabun*, instead, are vertical ovens [2]. The *tannur* consists of a truncated conical structure (Figure 8).

Archeological remains of these ovens are widespread in the Middle East, Central Asia, northern India, North Africa, and along the Mediterranean coasts [2]. The *tannur* is still used in the rural areas of the Middle East, particularly in Syria and Iraq, whereas it disappeared in Egypt, where it was used until the 19th century [64].

Embedded in the masonry, which acts as a workbench to prepare the dough and lay the bread loaves for cooling, the *tannur* is placed in a slightly inclined position to facilitate the introduction of food to be baked, including bread, through the circular opening at the top, or “mouth” [2]. The dough discs are rapidly pressed onto the inner walls of *tannur* with the aid of a “bread cushion” or directly by hand. The adhesion to the vertical walls can be helped by wetting the surface of the dough discs just before slapping them into the oven [2]. After about 1–2 min of cooking, the bread is taken out by metal tongs. The vertical ovens, such as all wood-burning ovens, reach very high temperatures, exceeding 300 °C.

The *tabun*, instead, has an upper opening as the *tannur*, but has an “igloo” shape, wider than tall [2] (Figure 9). This kind of oven, of Palestinian origin, is present in Jordan also due to large presence in this country of Palestinian refugees [46,65,66,67]. It is used in a different way than *tannur* because, instead of slapping onto the inner walls, the loaves are placed on the oven floor, next to the embers, usually on a layer of hot pebbles [2]. During baking, the top opening of the oven is closed with a metal lid.

The wood-fired refractory stone oven, dome-shaped, was present in all the surveyed countries, but with a different “perception”. In countries such as Italy, for instance, the presence of a wood-fired refractory stone oven (“*forno a legna*”) in a *pizzeria* is perceived very positively and attracts customers.

On the contrary, the traditional “*furn fallahi*” (farmer oven) used to bake the *Bataw* bread in the rural areas of Egypt is perceived as obsolete and has been almost abandoned (although it will likely make a comeback due to the sharp rise in the price of gas following the current Ukraine crisis).

### 3.6. Bread Characteristics

Regarding size and shape (Table 8), 50 breads had a diameter between 10 and 40 cm, while seven breads were larger than 40 cm: the Italian *Borlengo* and *Testarolo*, the Croatian *Poljički soparnik*, the Jordan *Saaj* (or *Shrak*) and *Mashrouh*, the Lebanese *Saj* (or *Markouk*) and *Khobz*. Only one bread was smaller than 10 cm, namely the Italian *Crescentina di Modena*. Not all flat breads are circular: 34.2% of them were oval or rectangular.

An important quality characteristic of flat breads was golden color (45.4%). Moreover, texture was relevant, which should be crunchy (12.2%) for the hard flat bread types and soft (14.1%) for those pliable and rollable (Table 9).

In the marketing classification, bread is included in the group of products of frequently purchased products, characterized by short shelf life and subject to high risk of waste [15,68,69]. The shelf life of flat breads ranged from shorter than 3 days (58%), between 3 and 7 days (27.3%) and up to one year (9.1%) in case of hard, dry flat breads (Table 10). Hard breads were obtained by means of a two-step baking: the first to cook and the second to dry them. Traditionally, this procedure was typical of breads to be carried during the transhumance of sheep [2]. These flat breads were the Croatian *Mlinci* (meaning “Mills”) and *Zagorski mlinci*, the Egyptian *Bataw* and *Merahrah*, the Greek *Fylla Perek*, the Italian *Pane Carasau*, *Schuttelbrot*, *Guttiau*, *Pistoccu*, *Zichi* and *Puccia ladina*, the Lebanese *Mullat al smeed* and the Spanish *Torta Cenceña* (or *Torta de Gazpacho Manchego*). The latter only shares its name with the cold vegetable soup named *gazpacho* consumed during the summer mainly in the Andalusia region. *Gazpacho Manchego*, indeed, is a game meat stew from the Castilla La Mancha region, traditionally eaten with unleavened bread cakes. Its most peculiar ingredient is the unleavened bread cake (*Torta Cenceña*) which, originally, was the plate for *Gazpacho Manchego*.

In 5.6% of cases the shelf life was not specified because the product was prepared at a very small scale level and marketed unpackaged.

Studies showed that modified atmosphere packaging (40% carbon dioxide and 60% nitrogen), coupled with an oxygen absorbent sachet, prolonged the shelf life of *pita* bread up to 28 days [70]. Alternatively, sodium propionate (0.3%) can be added [71]. Innovative solutions for extending the shelf life of bread are under study, such as ethanol and/or essential oil emitters, antimicrobial films, nanopackaging, biodegradable and renewable packaging, and edible coatings [72].

### 3.7. Artisanal vs. Industrial Breads

Typically, the surveyed flat breads showed an artisanal character (82%) (Figure 10). Those produced at industrial level accounted for 7%, and were: *Piadina* (Italy), *Kmaj* (Jordan), *Mlinci* and *Kukuruzna miješana ciabatta* (Croatia), *Souvlakopita* (*Pita* bread) (Greece), *Baladi* and *Shami* bread (Egypt), *Khobz* (Arabic bread) (Lebanon). Another 11% was produced either way.

The Egyptian *Bataw*, which was the most traditional farmer bread in Egypt, can be considered a case study. It was not standardized, being made in a different way in different places. Farmers could prepare *Bataw* bread with wheat, corn, or mixture of these flours (the most common option), with or without fenugreek. Furthermore, this kind of bread was produced either in soft form or, to prolong shelf life, in hard, dry form.

These findings demonstrate that the formulation of *Bataw* bread, a traditional product made mostly on a family basis, depends on the local availability of raw materials and personal preference of the family. However, this bread is currently not as widely consumed as in the past, mostly due to the subsidization of another bread, namely *Baladi* bread. It should be considered that in Egypt, bread consumption is one of the highest in the world, so bread subsidies have a strong influence on consumer choice [34]. Likely, changes in lifestyle and increasing urbanization may further enhance the abandonment of *Bataw* breadmaking. Therefore, there is a concrete risk of losing the memory and knowledge behind this kind of bread. This happens continuously to many food products, everywhere, but since bread is much more than simply a nutritious food, being linked to identity and local knowledge [3], abandoning a certain type of artisanal bread, in favor of a more industrialized one, is a phenomenon with important cultural repercussions.

Besides the *Bataw*, other breads which were found to be made on a family basis were the Egyptian *Zallut*, *Khobz min el dorra al rafi’ah, Farasheeh,* and *Merahrah*; the Jordan *Arbood*; the Lebanese *Mishtah el jreesh* and *Mullat al smeed*; the Italian *Carchiola*, *Puccia alla spasa* and *Pitt’ajima*; the Greek *Lambropsomo*, *Christopsomo*, *Spargana tou Christou* and *Vasilopita trifti* (whose cultural importance and ethnographical aspects are highlighted in the next paragraph).

### 3.8. Specific Consumption Patterns

The analysis of data also provided a cultural view of flat breads within each country. Preferences for food, including bread, are influenced by culture, family habits, traditions, religious believes, and income [68]. One of the characteristics of contemporary consumer behavior and habits is that people often purchase products (including food products) not because they are “used for something”, but because they symbolize something, as described by Solomon et al. [73] and Jones [74]. The role a product plays in people’s lives, indeed, extends beyond the practical functions it fulfils. Bread is a “soul food” and is the symbolic food *par excellence*, to be treated with the respect due to what it represents.

A strong link has been observed between the consumption of a specific flat bread type and certain periods of the year having a religious meaning, highlighting the iconic nature of cooking and eating [74]. The Greek *Lambropsomo* (Easter Bread) and *Lagana* are related to Easter [16]. The *Lambropsomo* is prepared by intertwining three long cylinders of dough, symbolizing the Holy Trinity, and on the surface of the dough four hard-boiled red-colored eggs are placed, which remind the blood of Jesus Christ. The *Lagana* is specifically consumed on Clean Monday (*Kathari Deutera*), i.e., on the first day of Lent of the Eastern Christianity, when sinful attitudes and non-fasting foods are left behind. That day is also named “Ash Monday”, by analogy with Ash Wednesday (the day when the Western Churches begin Lent). In Greece, the *Christopsomo* is the typical Christmas Bread, eaten on Christmas’ eve. The surface of this bread is decorated with a cross of dough and walnuts in shell [16]. The latter symbolizes rebirth and are believed to bring prosperity to the family. Moreover, the Greek *Spargana tou Christou* is related to Christmas. The name of these very thin pancakes, prepared during Christmas lent, literally means “The swaddling clothes of Jesus” [16]. The *Vasilopita trifti* or “New year’s bread”, instead, is a traditional Greek bread served at midnight on New Year’s Eve to celebrate the life of Saint Basil of Caesarea (*Agios Vasilis*), who is Santa Claus according to Greek Christmas traditions [16]. After baking, a coin is inserted through the base of the bread, and whoever finds it is said to be granted luck for the rest of the year.

In Croatia, *Zagorski mlinci* are traditionally eaten with turkey at Christmas and New Year. Moreover, *Poljički soparnik* is known as the “poor man’s dish” because it was always prepared on fasting days (All Saints’ day or Good Friday) and holidays such as Christmas.

The Lebanese *Mishtah el jreesh*, now become a rare bread prepared only at home, is particularly associated with the Muslim month of Ramadan, for breaking the fast, while the *Mullat al smeed*, another rare Lebanese bread, was traditionally brought during the *Hajj* (pilgrimage to Mecca) because of its long shelf life (being dried) but, as travel times became shorter, *Mullat al smeed* began to disappear [75]. The meat variant of *Manouche*, instead, called *Lahem b aajin* (or “meat in dough”), is a “go to” meal for funerals and condolences in certain regions of Lebanon.

In Italy, the Sardinian double-layered flat bread *Spianata* was traditionally prepared in a decorated version for weddings, by using a special stamp named *pintadera*. The Italian *Borlengo* was typically consumed at Carnival, so its name is related to the Italian word “*burla*”, meaning “joke” [16]. The *Pizza a scannatur di Carbone*, instead, was prepared on the day of the pig slaughter which, in the past, represented an important day, to be celebrated as a collective rite ending with a common lunch for those who helped in the slaughter [16].

### 3.9. Promoting the Tradition

The application of quality schemes to flat breads deserves a specific discussion. As a way for promoting the most traditional and artisanal foods, several quality marks have been set up. This action is aimed at keeping their knowledge alive, reducing the erosion of food diversity, which is a modern problem induced by the globalization of food products.

The EU Regulation No 1151/2012 [20] provides the legal framework on quality schemes to protect the name of food and beverages having unique characteristics linked to their geographical origin as well as traditional know-how [20]. These quality schemes include the PDO, granted to products whose production, processing, and preparation are entirely made within a particular geographical area; the Protected Geographical Indication (PGI), awarded to products for which at least one of the stages of production, processing or preparation takes place within a particular geographical area; the TSG, for products whose quality is not linked to a specific geographical area but is based on traditional processing methods or recipes. These quality marks are all recognizable by the presence of specific logos in the label of food and bring benefits to producers, who keep the exclusive right to use the protected name and usually get a premium price. At the same time, the consumers have a proof of heritage and tradition of food specialties they buy.

Many of the surveyed flat breads have been awarded of quality marks (Table 11). Five were PGI (namely the Italian *Piadina Romagnola*, *Focaccia di Recco* and *Schuttelbrot dell’Alto Adige*, as well as the Croatian *Poljički soparnik’* and *Zagorski mlinci*) and one was TSG (the *Pizza Napoletana*). Furthermore, being the “art” of the Neapolitan pizza-makers (*pizzaiuoli napoletani*) globally renowned, it has been inscribed on the Representative List of the Intangible Cultural Heritage of Humanity by the United Nations Educational, Scientific and Cultural Organization (UNESCO). Similarly, “the culinary art and culture of flattened sourdough flat bread *Ftira”*, was added by the UNESCO to the same list, being a key part of the cultural heritage of the inhabitants of the Maltese archipelago.

According to the Slow Food Foundation for Biodiversity, Slow Food Presidia should be good tasting, sustainably produced, should have a local and social dimension, and represent a sense of place. This recognition requires that producers join a project to safeguard biodiversity and form a community. Two flat breads of this survey were Slow Food presidia, i.e., the Italian *Testarolo Pontremolese* and the Spanish *Talo*.

In Croatia, the quality schemes “Intangible cultural goods” (awarded by the Ministry of Culture, of the Republic of Croatia), and “Croatian quality” (by the Croatian Chamber of Commerce) applied to *Pogača z oreji* and *Pogača*, respectively.

The “*Denominazione Comunale di Origine*” (DeCO, meaning “Municipal Designation of Origin”), instead, is granted by the Italian Municipalities to recognize, promote, and protect high quality artisanal agri-food products indigenous to their municipal territory. Two Italian breads were DeCO, while the remarkable number of 63 were *Prodotti Agroalimentari Tradizionali* (PAT, meaning “Traditional Agri-food Products”), which is another (more prestigious) Italian recognition awarded by the Italian Ministry of Agriculture to foods prepared according to traditional processing systems, homogeneous in the geographic area and consolidated through a period of time not inferior to 25 years (Table 12).

A list of PAT is released yearly, with new entries and eventually deletions for products which have been upgraded to PDO, PGI, or TSG [76]. PAT flat breads were homogeneously distributed throughout the country.

Rare breads, instead, prepared only by the household cooks for family consumption and not for sale, were surveyed by the Slow Food Foundation for Biodiversity within the “Ark of Taste” project. These breads were the Egyptian *Zallut*, *Khobz min el dorra al rafi’ah, Farasheeh,* and *Merahrah* [77], the Lebanese *Mishtah el jreesh* and *Mullat al smeed* [75], the Jordan *Arbood* [78] and the Italian breads reported in Table 13. The Ark of Taste project is aimed at keeping alive small-scale quality productions tightly linked to local culture, history and tradition [79,80].

It is indeed fundamental to draw attention to these local rare foods which are at risk of extinction, to prevent the reduction of food and cultural biodiversity. The Egyptian *Shamsi* bread, for example, which was another rare bread, has recently increased its popularity and has now gained certain market, as proved by the large number of webpages released by searching its name in Google. A possible means for keeping alive these food products is also to show their production to the tourists, who are attracted by local traditions, such as for the Jordan *Arbood* bread. This bread is typically baked for tourists enjoying the Wadi Rum tours. In this way, the promotion of traditional foods can generate an income to the local communities [81].

Another means to enhance the knowledge of rare flat breads is represented by the food blogs which attract the interest of many people prompt by the will of learning new recipes and rediscovering old ones. These tools, as well as YouTube videos, or even Wikipedia, easily carry the information also far from the area of origin. For example, the Italian breads inserted in the Ark of Taste (namely *Carchiola, Pizza scime—or Pizza scive, Pizza ascima, Pizza azzima—Pitt’ajima, Pizza a fiamma, Puccia alla spasa* and *Puccia ladina*) are all the object of several blogs and are mentioned in Wikipedia pages, with *Carchiola* having its own entry.

In Lebanon, instead, the popular garnished flat bread *Manouche* is the object of the “World *Manoucheh* Day”, on November 2nd, though this bread has not an official quality label yet.

## 4. Conclusions

This survey provides an overview of the different recipe, process, and product quality specifications of flat breads produced in the examined countries, as well as an insight on the related local culture. It appears that flat breads are really a large and diverse family of food products, with a rich history and ethnographical dimension.

Their main technical characteristics could be summarized as follows. Flours were from soft wheat (67.4%), durum wheat (13.7%), corn (8.6%), rye, sorghum, chickpea, and chestnut (together 5.2%). All garnished flat breads contained vegetable oils or lard, while 54% of plain flat breads did not contain fats. Leavening agents were: compressed yeast (55.8%), sourdough (16.7%), or baking powder (9.0%); 18.6% flat breads were unleavened. The baking temperatures were <250 °C (50.3%), between 250 and 300 °C (7%), and >300 °C (28.7%). Sixteen old-style baking systems were recorded, classified into baking plates and vertical ovens (*tannur* and *tabun*). Artisanal flat breads accounted for 82%, while the industrial for 7%. The diameter of breads was between 10 and 40 cm (39.9%), <10 cm (1%), >40 cm (4.9%). Not all flat breads were circular: 34.2% of them were oval or rectangular. The shelf life ranged from shorter than 3 days (58%), between 3 and 7 days (27.3%) and, in case of hard, dry flat breads, up to one year (9.1%). The main quality characteristics were golden color (45.4%), crunchiness (12.2%), and softness (14.1%) for hard and rollable flat breads, respectively. Quality schemes (national, European or global) applied to 91 flat breads.

In addition to the technical aspects, a clear social, ethnographical, and cultural dimension was identified. Twelve flat breads were strongly associated to religious celebrations, and 15 flat breads were rare, prepared only by the household cooks in a rural environment, for family consumption.

The collected information, gathered in a publicly available database, will be fundamental for allowing further valorization and dissemination activities, and will be useful for the selection, within each one of the examined country, of the most suitable flat breads for nutritional fortification and technical innovation within the FlatBreadMine or any other research project.

Criteria for selecting breads can be proposed, as follows:

(1) The type of bread chosen must be native to the country and widely consumed throughout the national territory. To maximize the nutritional impact on the general population, it would be of little usefulness to fortify a flat bread produced on a small-scale, consumed only in a restricted geographical area.

(2) If different categories of flat bread are available in the considered country, all equally diffused, more than one bread should be selected, to represent the single-layered, double-layered, and garnished or fried products.

## 5. Future Perspectives

Traditional and rare flat breads, prepared in a genuine and not globalized way using local raw materials, in addition to not risking the shortage of imported flours for geopolitical or climatic reasons, are well adapted to the territory and sustainable. However, these breads may progressively disappear due to changes in lifestyle and increasing urbanization, and their loss would lead to genetic erosion. Actions are therefore needed to prevent the reduction of cultural and biological diversity related to their disappearance. The strategy for such preventing actions, and directions for future research, should involve: (1) Periodically surveying the existing flat bread diversity, with the same approach used in this article. It is worth noting that the FlatBreadMine database can be easily updated, as well as extended to include many other countries; (2) applying quality schemes to the most genuine and high quality products; (3) disseminating information and raising awareness by promoting the products both remotely (food blogs, Youtube video) and in person, also among tourists (because food has to be tasted, to see it online is not enough).

Traditional flat breads produced on a small-scale and those, more industrialized, on a large-scale, have totally different features and should follow distinct paths, with the first as a specialty dedicated to a niche consumer, carrying a strong cultural message, and the latter for mass consumption, able to guarantee a real impact on the population if nutritionally improved.

## Figures and Tables

**Figure 1 foods-11-02326-f001:**
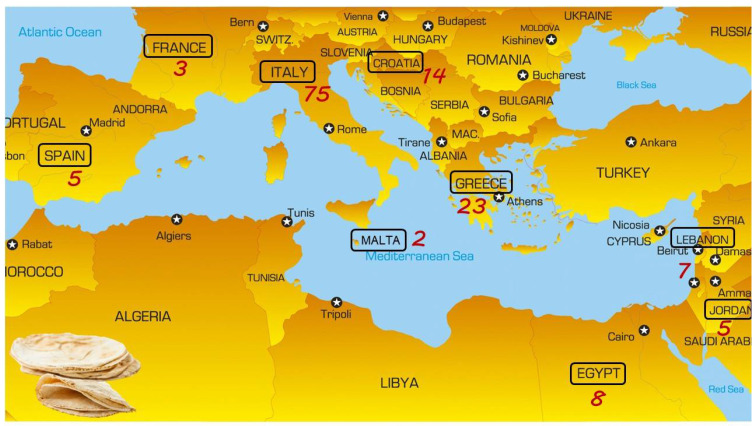
Geographical distribution of flat breads in the surveyed area (Croatia, Egypt, France, Greece, Italy, Jordan, Lebanon, Malta, and Spain).

**Figure 2 foods-11-02326-f002:**
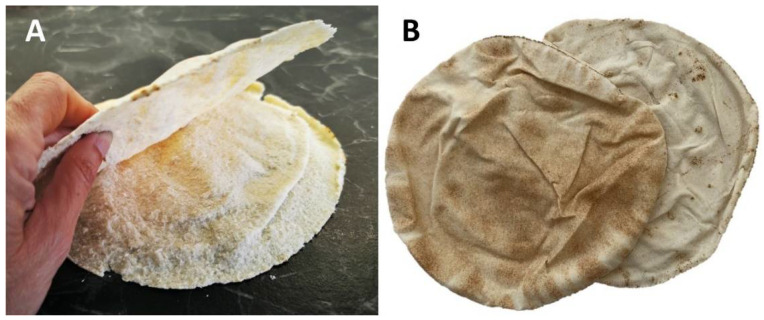
(**A**) Jordan *Kmaj* opened to show the internal “pocket”. (**B**) Lebanese *Khobz*.

**Figure 3 foods-11-02326-f003:**
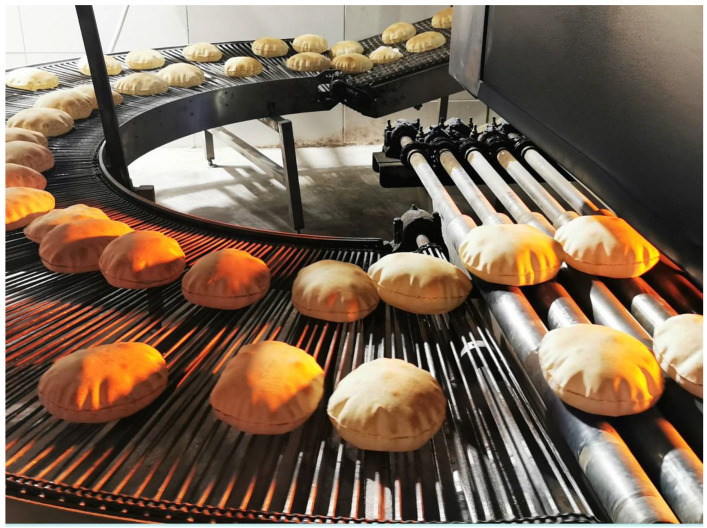
Jordan *Kmaj* in an automatic baking line. The inflation of bread due to the thermal expansion of the gases is clearly visible.

**Figure 4 foods-11-02326-f004:**
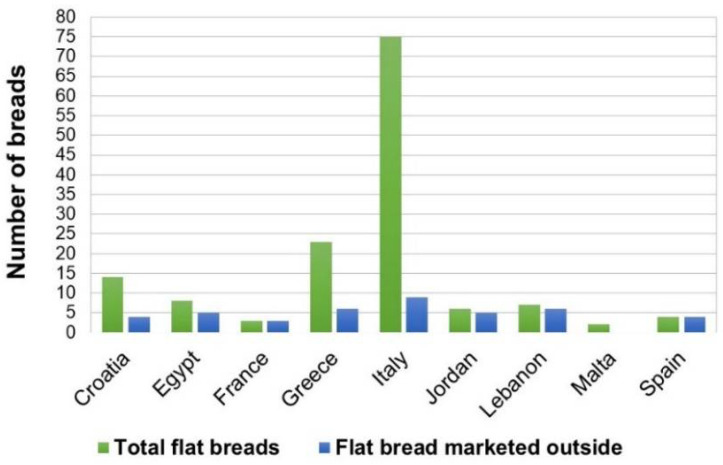
Number of flat bread types marketed outside the area of origin, compared to total flat breads per country (*n* = 143).

**Figure 5 foods-11-02326-f005:**
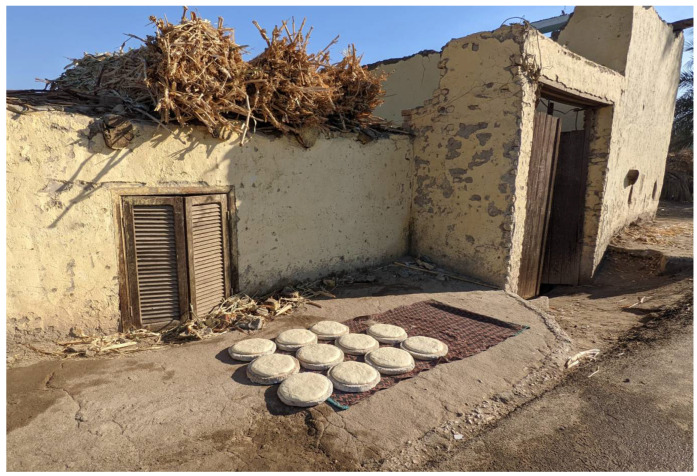
Preparation of Egyptian *Shamsi* bread: dough exposed to the sun for fermentation.

**Figure 6 foods-11-02326-f006:**
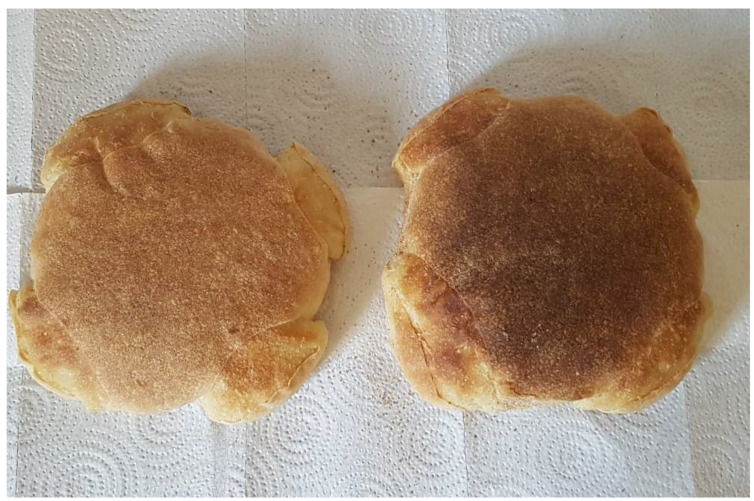
Egyptian *Shamsi* bread.

**Figure 7 foods-11-02326-f007:**
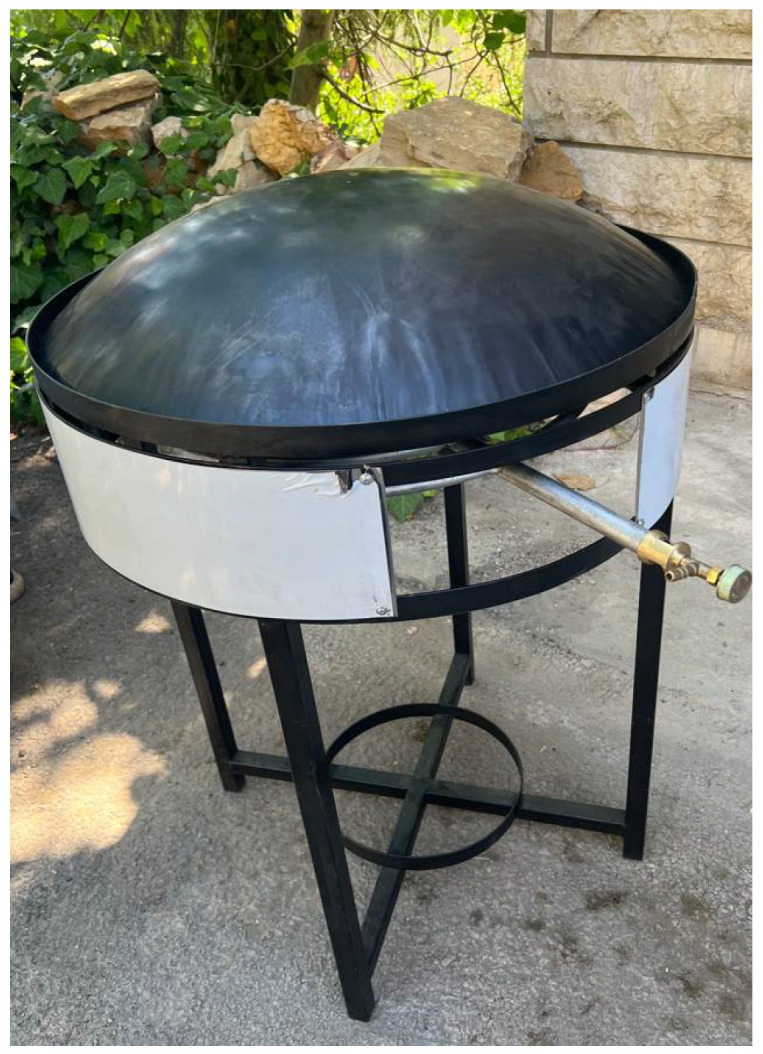
A domestic *saj*.

**Figure 8 foods-11-02326-f008:**
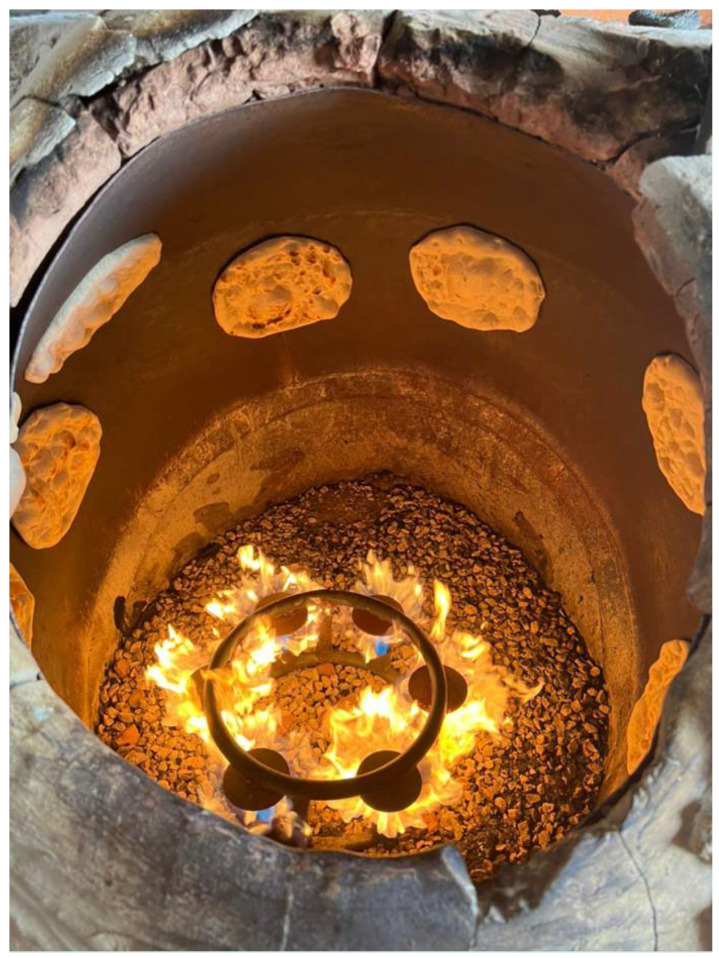
Dough discs pressed onto the inner walls of *tannur* for baking.

**Figure 9 foods-11-02326-f009:**
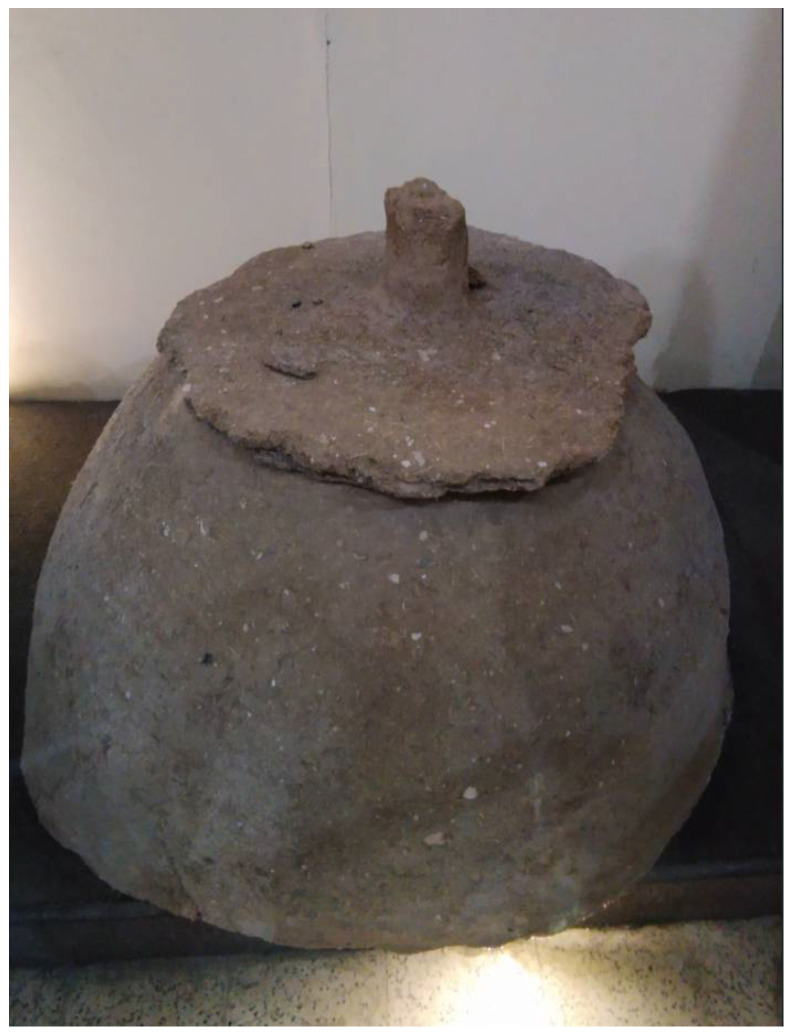
*Tabun* oven.

**Figure 10 foods-11-02326-f010:**
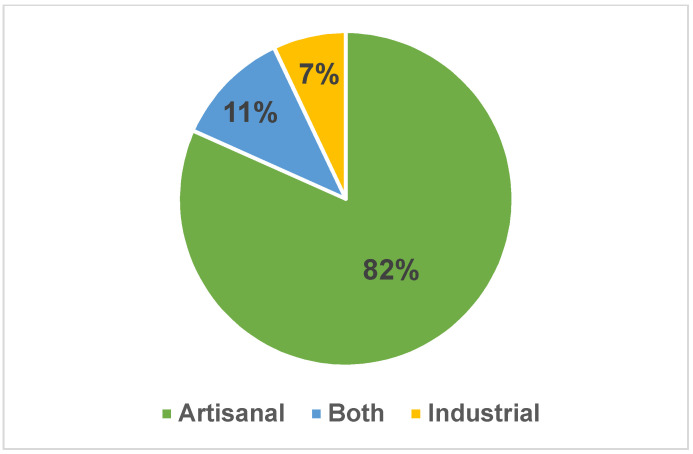
Percentage of flat breads produced in artisanal or industrial mode.

**Table 1 foods-11-02326-t001:** Occurrence of flat breads in the different categories.

Country	Flat Bread Category							
	Plain				Garnished (Seasoned, Stuffed)		Fried	
	Single-Layered		Double-Layered					
	Number	%	Number	%	Number	%	Number	%
Croatia	5	3.5	2	1.4	7	4.9	-	-
Egypt	6	4.2	2	1.4	-	-	-	-
France	-	-	-	-	3	2.1	-	-
Greece	6	4.2	1	0.7	12	8.4	4	2.8
Italy	23	16.1	7	4.9	39	27.3	6	4.2
Jordan	5	3.5	1	0.7	-	-	-	-
Lebanon	3	2.1	1	0.7	3	2.1	-	-
Malta	-	-	1	0.7	1	0.7	-	-
Spain	3	2.1	-	-	1	0.7	1	0.7
Total	51	35.7	15	10.5	66	46.2	11	7.7

**Table 2 foods-11-02326-t002:** Type of flour used in the preparation of flat breads. Multiple answers were admitted, because breads could be prepared with different flours or with flour blends.

Country	Type of Flour											
	Soft Wheat				Durum Wheat ^a^		Corn ^a^		Rye ^a^		Other Species ^a,b^	
	Refined		Whole Meal									
	Number	%	Number	%	Number	%	Number	%	Number	%	Number	%
Croatia	14	8.0	2	1.1	-	-	3	1.7	2	1.1	-	-
Egypt	5	2.9	1	0.6	-	-	2	0.6	-	-	2	1.1
France	3	1.7	-	-	-	-	-	-	-	-	-	-
Greece	20	11.4	2	1.1	4 ^c^	-	3	1.7	-	-	-	-
Italy	59	33.7	2	1.1	14	8.0	5	2.9	2	1.1	3	1.7
Jordan	6	3.4	-	-	5	2.9	-	-	-	-	-	-
Lebanon	7	4.0	1	0.6	1	0.6	1	0.6	-	-	-	-
Malta	2	1.1	1	0.6	-	-	-	-	-	-	-	-
Spain	3	1.7	-	-	-	-	2	1.1	-	-	-	-
Total	118	67.4	9	5.1	24	13.7	16	8.6	4	2.3	5	2.9

^a^ Refined flour, unless otherwise specified. ^b^ Sorghum, chickpea, chestnut. ^c^ Durum wheat whole meal is used in the preparation of *Koulouri* (Greece).

**Table 3 foods-11-02326-t003:** Type of fat eventually used in plain flat breads. Multiple answers were admitted, because breads could be prepared with different oils and fats or with blends.

Country	Olive Oil		Sunflower Oil		Vegetable Oil (Not Specified)		Lard		None	
	Number	%	Number	%	Number	%	Number	%	Number	%
Croatia	1	1.4	2	2.9	1	1.4	-	-	3	4.3
Egypt	-	-	-	-	1	1.4	-	-	7	10.0
France	-	-	-	-	-	-	-	-	-	-
Greece	2	2.9	-	-	-	-	-	-	5	7.1
Italy	13 ^a^	18.6	-	-	-	-	8	11.4	12	17.1
Jordan	-	-	-	-	-	-	-	-	6	8.6
Lebanon	1	1.4	-	-	1	1.4	-	-	3	4.3
Malta	1 ^a^	1.4	-	-	-	-	-	-	-	-
Spain	1	1.4	-	-	-	-	-	-	2	2.9
Total	19	27.1	2	2.9	3	4.3	8	11.4	38	54.3

^a^ Extra virgin olive oil is used in nine Italian flat breads and in the Maltese one.

**Table 4 foods-11-02326-t004:** Characteristic ingredients of garnished flat breads. Multiple answers were admitted, because breads could be prepared with different fats and ingredients.

Country	Fats				Additional Ingredients											
	Vegetable Oil ^a^		Lard		Plant-Based Ingredients ^b^		Dairy Products		Eggs		Cured Meat		Canned Fish		Meat	
	Number	%	Number	%	Number	%	Number	%	Number	%	Number	%	Number	%	Number	%
Croatia	5	8.3	2	3.3	7	6.3	3	2.7	2	1.8	-	-	3	2.7	-	-
Egypt	-	-	-	-	-	-	-	-	-	-	-	-	-	-	-	-
France	3	5.0	-	-	2	1.8	1	0.9	-	-	1	0.9	1	0.9	-	-
Greece	10	16.7	-	-	9	8.1	5	4.5	6	5.4	-	-	-	-	-	-
Italy	28	46.7	10	16.7	23	20.7	12	10.8	5	4.5	15	13.5	6	5.4	1	0.9
Jordan	-	-	-	-	-	-	-	-	-	-	-	-	-	-	-	-
Lebanon	1	1.7	-	-	3	2.7	2	1.8	-	-	-	-	-	-	1	0.9
Malta	-	-	-	-	1	0.9	1	0.9	1	0.9	-	-	-	-	-	-
Spain	1	1.7	-	-	-	-	-	-	-	-	-	-	-	-	-	-
Total	48	80.0	12	20.0	45	40.5	24	21.6	14	12.6	16	14.4	10	9.0	2	1.8

^a^ When specified, olive oil, sunflower oil, rapeseed oil, or blends of these were used. ^b^ Spices, vegetables, cereals, seeds.

**Table 5 foods-11-02326-t005:** Type of yeast, if any, used in the preparation of flat breads. Multiple answers were admitted, because breads could be prepared with different types of yeast or yeast mixtures.

Country	Type of Yeast							
	Baking Powder		Compressed Yeast		Sourdough		None	
	Number	%	Number	%	Number	%	Number	%
Croatia	2	1.3	9	5.8	2	1.3	3	1.9
Egypt	1	0.6	5	3.2	1	0.6	-	-
France	-	-	3	1.9	-	-	-	-
Greece	1	0.6	17	10.9	1	0.6	5	3.2
Italy	7	4.5	40	25.6	19	12.2	16	10.3
Jordan	3	1.9	4	2.6	-	-	1	0.6
Lebanon	-	-	6	3.8	-	-	1	0.6
Malta	-	-	1	0.6	2	1.3	-	-
Spain	-	-	2	1.3	1	0.6	3	1.9
Total	14	9.0	87	55.8	26	16.7	29	18.6

**Table 6 foods-11-02326-t006:** Baking temperature adopted in the baking process of flat breads.

Country	Temperature (°C)							
	<250		250–300		>300		Not Specified	
	Number	%	Number	%	Number	%	Number	%
Croatia	7	4.9	1	0.7	2	1.4	4	2.8
Egypt	-	-	1	0.7	7	4.9	-	-
France	3	2.1	-	-	-	-	-	-
Greece	13	9.1	-	-	2	1.4	8	5.6
Italy	48	33.6	4	2.8	22	15.4	1	0.7
Jordan	-	-	1	0.7	4	2.8	1	0.7
Lebanon	1	0.7	3	2.1	3	2.1	-	-
Malta	-	-	-	-	1	0.7	1	0.7
Spain	-	-	-	-	-	-	5	3.5
Total	72	50.3	10	7.0	41	28.7	20	14.0

**Table 7 foods-11-02326-t007:** Traditional baking systems used in the preparation of flat breads.

Traditional Baking System	Country and Breads
On a hot sandy ground, under hot ashes and embers	Jordan: *Arbood*
Metal grill (*R’ticula*) on embers	Italy: *Carchiola, Crostolo*
In the fireplace (named *Komin* in Croatia, *Camino* in Italy), under hot ashes and embers	Croatia: *Poljički soparnik*; Italy: *Crescia sotto la cenere*
On the hearth, under a bell-shaped iron lid (*Peka*) covered with embers	Croatia: *Kruh ispod peke* (‘‘Bread under the lid’’ or “The *Peka*”)
In the fireplace, under a terracotta lid (*Coppo*) covered with embers	Italy: *Pizza scime (or Pizza scive, Pizza ascima, Pizza azzima); Pizza somma*
Baking stone	Greece: *Plakopita*, *Spargana tou Christou*
Terracotta plate (*Tégia*) ^a^	Italy, *Piadina Romagnola*
Terracotta plate, smaller than *Tégia* (*Tigella*) ^b^	Italy, *Crescentina di Modena*
Terracotta plate with a terracotta lid (*Testo*) ^a^	Italy: *Testarolo Pontremolese*, *Panigaccio* ^c^*, Torta al Testo* ^d^, *Neccio* ^d^, *Crescia sfogliata* ^d^
Iron griddle	Spain: *Talo*
Circular convex metal griddle (*Saj* or, only in Greece, *Satsi*)	Jordan: *Mashrouh, Saaj*; Lebanon: *Saj;* Egypt: *Farasheeh*; Greece: *Fylla Perek (Perek sheets)*
Metal pan	Greece: *Pisia, Tiganopsomo*, *Sfakianopita, Fyllota* Italy: *Farinata*
Tinned copper pan (*Sole*)	Italy: *Borlengo di Guiglia*
Igloo-shaped clay oven (*Tabun*)	Jordan: *Tabun*
Vertical, tubular-shaped clay oven (*Tannur*)	Jordan: *Tannur*; Lebanon: *Tannur*
Refractory stone oven, dome-shaped	All surveyed countries, with many breads each

^a^ Modern versions made of metal are currently used; ^b^ Modern versions of this cooking system consist of two flat cast iron discs, between which the dough is cooked; ^c^ *Panigaccio* is cooked without the lid, between two *testo* plates. Multiple *testo* plates can be stacked; ^d^ *Torta al Testo*, *Neccio* and *Crescia sfogliata* are cooked without the lid, but have to be flipped during baking to cook homogeneously on both sides.

**Table 8 foods-11-02326-t008:** Shape characteristics of flat breads.

Country	Circular (Diameter, cm)								Other Shapes ^a^	
	<10		10–40		>40		Not Specified			
	Number	%	Number	%	Number	%	Number	%	Number	%
Croatia	-	-	4	2.8	1	0.7	-	-	9	6.3
Egypt	-	-	5	3.5	-	-	3	2.1	-	-
France	-	-	-	-	-	-	-	-	3	2.1
Greece	-	-	18	12.6	-	-	-	-	5	3.5
Italy	1	0.7	17	11.9	2	1.4	28	19.6	27	18.9
Jordan	-	-	3	2.1	2	1.4	1	0.7	-	-
Lebanon	-	-	5	3.5	2	1.4	-	-	-	-
Malta	-	-	2	1.4	-	-	-	-	-	-
Spain	-	-	3	2.1	-	-	-	-	2	1.4
Total	1	0.7	57	39.9	7	4.9	32	22.4	46	32.2

^a^ Rectangular or oval.

**Table 9 foods-11-02326-t009:** Principal quality characteristics of flat breads. Multiple answers were admitted, because breads could show more quality characteristics at the same time.

Country	Quality Characteristics							
	Golden Color		Crunchy Texture		Soft Texture and/or Pliability		Not Specified	
	Number	%	Number	%	Number	%	Number	%
Croatia	13	8.6	2	1.2	1	0.6	-	-
Egypt	5	2.9	-	-	-	-	3	1.8
France	2	1.3	-	-	-	-	1	0.6
Greece	13	8.6	7	4.1	11	6.5	-	-
Italy	21	15.4	8	5.2	8	4.7	41	24.1
Jordan	5	2.9	-	-	-	-	1	0.6
Lebanon	6	3.6	1	0.6	3	1.8	-	-
Malta	1	0.7	2	1.2	1	0.6	-	-
Spain	2	1.3	-	-	-	-	2	1.2
Total	68	45.4	30	12.2	24	14.1	48	28.24

**Table 10 foods-11-02326-t010:** Shelf life of flat breads.

Country	Shelf Life							
	<3 Days		3–7 Days		Up to 1 Year ^a^		Not Specified	
	Number	%	Number	%	Number	%	Number	%
Croatia	3	2.1	1	0.7	2	1.4	8	5.6
Egypt	1	0.7	5	3.5	2	1.4	-	-
France	2	1.4	1	0.7	-	-	-	-
Greece	19	13.3	3	2.1	1	0.7	-	-
Italy	48	34.6	21	14.0	6	3.7	-	-
Jordan	5	3.5	1	-	-	-	-	-
Lebanon	3	2.1	3	2.1	1	0.7	-	-
Malta	2	1.4	-	-	-	-	-	-
Spain	-	-	4	2.8	1	0.7	-	-
Total	83	58.0	39	27.3	13	9.1	8	5.6

^a^ Dry breads.

**Table 11 foods-11-02326-t011:** Flat breads awarded by quality marks. PGI = Protected Geographical Indication; TSG = Traditional Specialty Guaranteed; PAT = *Prodotti agroalimentari tradizionali* (Traditional Agri-food Products); DeCO = *Denominazione Comunale di Origine* (Municipal Designation of Origin).

Quality Mark		Releasing Organism	Geographic Validity	Number of Breads	Bread Names and Country
Name	Acronym				
Protected Geographical Indication	PGI	European Commission	EU	5	*Piadina Romagnola* (Italy); *Schuttelbrot* (Italy); *Focaccia di Recco* (Italy); *Poljički soparnik’* (Croatia); *Zagorski mlinci* (Croatia)
Guaranteed Traditional Specialty	TSG	European Commission	EU	1	*Pizza Napoletana* (Italy)
Intangible cultural heritage of humanity	-	UNESCO	Global	2	The culinary art and culture of flattened sourdough flat bread *Ftira* (Malta); The Art of Neapolitan Pizza-makers (Italy)
Slow Food presidium	-	Slow Food Foundation for Biodiversity	Global	2	*Testarolo Pontremolese* (Italy); *Mungia Talo* (Spain)
Intangible cultural goods	-	Ministry of Culture, of the Republic of Croatia	Croatia	1	*Pogača z oreji* (Croatia)
Croatian Quality	-	Croatian Economy Chamber	Croatia	1	*Pogača Pogacha* (Croatia)
Municipal Designation of Origin	DeCO	Italian Municipalities	Italy	2	*Crostolo di Urbania* (Italy); *Farinata di Imperia* (Italy)
Traditional Agri-food Products	PAT	Italian Ministry of Agriculture, Food and Forestry	Italy	63	See detailed list in Table 12

**Table 12 foods-11-02326-t012:** Italian flat breads awarded by the Italian quality mark “*Prodotti Agroalimentari Tradizionali*” (PAT, meaning “Agri-food Traditional Products”), geographically subdivided based on their origin in northern, central, southern Italy or its islands.

Region	Number	Bread Names
Northern Italy		
Aosta Valley	-	-
Piedmont	2	*Farinata, Focaccia Novese*
Trentino-Alto Adige	-	-
Friuli-Venezia Giulia	-	-
Veneto	-	-
Lombardy	1	*Schiacciatina*
Emilia-Romagna	8	*Crostolo, Borlengo di Guiglia, Crescentina, Gnocco fritto, Crescenta fritta, Focaccia con ciccioli, Erbazzone, Crescenta*
Liguria	3	*Testarolo della Lunigiana, Farinata, Focaccia*
Total	14	*-*
Central Italy		
Tuscany	6	*Testaroli, Panigaccio, Neccio, Farinata, Schiaccia grossetana, Focaccia con i friccioli*
Umbria	2	*Torta al Testo, Schiacciata al formaggio*
Marches	5	*Crostolo, Cresciolina, Crescia sotto la cenere, Crescia d’la stacciola, Crescia brusca*
Abruzzo	2	*Pizza Scime (or Pizza scive, Pizza ascima, Pizza azzima), Pizza con le sfrigole*
Lazio	7	*Pizza bianca romana, Pizza con farina di mais, Pizza somma, Pizza rossa, Pizza fritta, Pizza sotto la brace, Pizza a fiamma*
Molise	3	*Pizza coi cicoli di maiale, Pizza di granone, Pizza scimia*
Total	25	*-*
Southern Italy		
Apulia	10	*Puccia salentina, Focaccia barese, Focaccia di S. Giuseppe, Calzone di Ischitella, Focaccia a libro di Sammichele di Bari, Paposcia, Pitilla, Pizza sfoglia, Scannatedda, Sceblasti*
Basilicata	4	*Carchiola, Pizza a scannatur di Carbone, Scarcedda, Pizza con i cingoli di maiale*
Calabria	1	*Pizza di maggio*
Campania	3	*Pizza, Pizza di farinella bacolese, Pizza di scarola*
Total	18	*-*
Islands		
Sardinia	6	*Carasau bread, Guttiau, Spianata, Pistoccu, Zichi, Focaccia Portoscusese*
Sicily	1	*Sfincione*
Total	7	*-*

**Table 13 foods-11-02326-t013:** Breads surveyed by Slow Food Foundation for Biodiversity as “Ark of Taste”.

Country	Number	Bread Names
Egypt	5	*Khobz min el dorra al rafi’ah, Farasheeh, Zallut, Merahrah, Shamsi*
Italy	7	*Carchiola, Pizza scime (or Pizza scive, Pizza ascima, Pizza azzima), Pitt’ajima, Pizza a fiamma, Puccia alla spasa, Puccia ladina, Focaccia a libro di Sammichele*
Jordan	1	*Arbood*
Lebanon	2	*Mishtah, Mullat al smeed*
Total	15	*-*

## Data Availability

The database is publicly accessible at the link: https://flatbreadmine.eu/resources/ (accessed on 7 July 2022).

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
