# Peer review of "The Large and Diverse Family of Mediterranean Flat Breads: A Database"

_foods, 2022, doi:10.3390/foods11152326_

Round 1
Reviewer 1 Report
The manuscript ‘The Large and Diverse Family of Mediterranean Flat Breads: A Database’ is a study to identify the main characteristics and diversity among flat breads produced in selected Mediterranean countries. The study is very interesting for a potential reader and important for the community. The paper is nicely readable and is worth to be presented but the paper needs improvements:
COMMENTS:
- Abstract should be rewritten. Authors should keep the following scheme: motivation, characterization of data, conclusions. Be precise. Avoid terms like ‘some were local..’ (l. 37-38), etc. The main findings presented in abstract should be a short resume of comclusions section
- Explain why other Meditt. countries like Totkey, Algeria, Libia, Morocco were omitted from the analysis. The justification of motivation and reliability of the results is needed
- The introduction should be extended by a section devoted to overview of flat breads produced all over the world (Asia, Americas. It seems like only the Meditt. European countries produced such bread.
- explain the reason of removing some categories from the survey (l. 113) like pizza – the wat of bread is different for different countries. Explain how can you avoid misleading conclusions because of such limitations.
- The collected data file looks impressive but is hard to analyze by the reader in the pdf format (l. 126). The data should available in excel format.
- Results obtained from different sources are collected in on place, but no statistics are present. Results are presented as a real number for different features, but can lead to wrong conclusions by the comparison uncomparable results without statistical significance. It is unreasonable to claim that some features are better, shorter, etc. Reformulation of sentences or additional information regarding such threat is needed
- Figure 10 is unreadable. Should be corrected.
- The conclusions section should be rewritten. The main findings as declared in the paper (l. 98) regarding: i) the regional area or town of origin, and the area of marketing and diffusion; ii) the ingredients used in bread preparation (flour, yeast, additional ingredients and their ratio); iii) the raw material characteristics; iv) the production process, v) the characteristics of bread should be placed. The conclusion should be clear, precise and understandable by the reader. There is no meaning of conclusions like ‘some flat breads … are prepared only by households …, which in some cases …’. Find patterns, or identify main differences.
Author Response
Reviewer 1
The manuscript ‘The Large and Diverse Family of Mediterranean Flat Breads: A Database’ is a study to identify the main characteristics and diversity among flat breads produced in selected Mediterranean countries. The study is very interesting for a potential reader and important for the community. The paper is nicely readable and is worth to be presented but the paper needs improvements:
Answer: We thank the reviewer for appreciating our work and for helpful suggestions. All the points raised have been considered. Modifications have been made in red text.
COMMENTS:
- Abstract should be rewritten. Authors should keep the following scheme: motivation, characterization of data, conclusions. Be precise. Avoid terms like ‘some were local..’ (l. 37-38), etc. The main findings presented in abstract should be a short resume of comclusions section
Answer: We added the motivation (see lines 30-31, 39-40) and substituted the word “some” with the exact number of breads at line 37-38 (see line 37) and throughout the entire manuscript, where any generic word (several, usually, commonly) has been substituted by the exact number of breads and their names (see lines 202-204; 211-214; 514-518; 553-557). We made the abstract adhere more to the Conclusion section.
- Explain why other Meditt. countries like Totkey, Algeria, Libia, Morocco were omitted from the analysis. The justification of motivation and reliability of the results is needed
Answer: We better explained the reason for focusing only on nine Mediterranean countries (see lines 29-30; 97-99). The Reviewer will agree that such a wide research needs to be financed, and the research project involved only the nine countries mentioned in the paper. However, we consider this paper as a starting point for further expanding the research to other countries, hoping in further financing. We added the sentence “It is worth noting that the FlatBreadMine database could be easily updated, as well extended to include many other countries where flat breads are produced” Lines 755-756.
- The introduction should be extended by a section devoted to overview of flat breads produced all over the world (Asia, Americas. It seems like only the Meditt. European countries produced such bread.
Answer: Thanks for suggestion, we added a sentence to state clearly that flat breads are produced all over the world (Asia, Americas) (see lines 49-54).
- explain the reason of removing some categories from the survey (l. 113) like pizza – the wat of bread is different for different countries. Explain how can you avoid misleading conclusions because of such limitations.
Answer: We did not remove pizza from the survey, sorry the sentence was unclear. The survey included all the flat breads of the examined countries, no one excluded. The strongest point of this work is the good extension of the dataset and the accuracy in the collection of data, made also with the help of experts in each country, to be sure of not excluding any product. We modified the sentence to make it clearer (see lines 113-115).
We also better clarified that the first step of this investigation was to identify all the flat bread types produced in each one of the Mediterranean countries involved in the FlatBreadMine Project (lines 126-127). The second step, after having identified all the flat breads, was to collect information on their ingredients, production process, etc. (Lines 135-136). Therefore, we did not select a group of flat breads per country, we collected info regarding all flat breads in each country.
- The collected data file looks impressive but is hard to analyze by the reader in the pdf format (l. 126). The data should available in excel format.
Answer. Sorry but, for avoiding manipulating the file, the most suitable format was pdf, as suggested by the webmaster of the project. However, the pdf file can be enlarged (up to 400%), becoming readable.
- Results obtained from different sources are collected in on place, but no statistics are present. Results are presented as a real number for different features, but can lead to wrong conclusions by the comparison uncomparable results without statistical significance. It is unreasonable to claim that some features are better, shorter, etc. Reformulation of sentences or additional information regarding such threat is needed
Answer: We reformulated all the sentences to avoid comparisons. See lines 195-198; 201-202; 211-214; 233; 345-346; 378-380; 417-418, 515-519; 528-531; 554-558
- Figure 10 is unreadable. Should be corrected.
Answer: We corrected the Fig. 10.
- The conclusions section should be rewritten. The main findings as declared in the paper (l. 98) regarding: i) the regional area or town of origin, and the area of marketing and diffusion; ii) the ingredients used in bread preparation (flour, yeast, additional ingredients and their ratio); iii) the raw material characteristics; iv) the production process, v) the characteristics of bread should be placed. The conclusion should be clear, precise and understandable by the reader. There is no meaning of conclusions like ‘some flat breads … are prepared only by households …, which in some cases …’. Find patterns, or identify main differences.
Answer: We modified the conclusion (which now are more similar to the abstract, as suggested) to summarize all the main findings (see lines 719-742), and we moved part of the previous Conclusions to a Future perspectives section, to highlight important considerations for future work.

Reviewer 2 Report
In my opinion, it is an interesting study which provides information which was not yet published and is of interest to readers. Additionally, it fulfilled Sustainable food system goals.
I suggest improving the Figures as most of them are of a low graphical quality.
Figure 4: Improve the quality and clarify the title “Diffusion of flat breads marketed outside the area of origin,…”
Line 264: explain the abbreviation PDO as it is for the first time mentioned in the manuscript.
In figure 10 the values are not visible
Line 488: are you sure that Spanish Gazpacho may reefers to bread because it is known as a cold tomato and other vegetable soup. Please clarify it.
All in all, it is an interesting and well-written study which fits well into the scope of the Foods journal.
Author Response
Reviewer 2
In my opinion, it is an interesting study which provides information which was not yet published and is of interest to readers. Additionally, it fulfilled Sustainable food system goals.
Answer: We thank the reviewer for appreciating our work. All the points raised have been considered. Modifications have been made in red text.
I suggest improving the Figures as most of them are of a low graphical quality.
Answer: We improved the figures.
Figure 4: Improve the quality and clarify the title “Diffusion of flat breads marketed outside the area of origin,…”
Answer: We improved the figure quality and clarified the title as suggested.
Line 264: explain the abbreviation PDO as it is for the first time mentioned in the manuscript.
Answer: The abbreviation of PDO was already explained at its first mentioning (see line 246, in green).
In figure 10 the values are not visible
Answer: We improved the figure quality.
Line 488: are you sure that Spanish Gazpacho may reefers to bread because it is known as a cold tomato and other vegetable soup. Please clarify it.
Answer: Actually Gazpacho referred to a dish named “Gazpacho Manchego”, i.e. a game meat stew from the Castilla La Mancha region, traditionally eaten with unleavened bread cakes (Torta Cenceña). The sentence was very unclear, we reworded all, improved naming and provided an explanation (see lines 537-542).
All in all, it is an interesting and well-written study which fits well into the scope of the Foods journal.
Answer: Thank you very much.

Reviewer 3 Report
Manuskrypt - Foods-1835792-
The Large and Diverse Family of Mediterranean Flat Breads: A Database
This article presents an interesting study on the richness of the bread market from a local perspective, which should be considered in line with the UN's sustainable development agenda. The local food market, with particular reference to the richness of a product group such as bread, represents a wide field of practice and requires work from a practical and scientific perspective. Organisations involved in developing such foods should continually look for ideas to develop wide-ranging cooperation, thinking about how to improve competitiveness within the framework of sustainable development, involving the integration and balancing of various levels, including economic, social, technical, environmental, political and natural. Research is needed to make an 'inventory of products as well as producers'. There should be action to prevent the reduction of cultural and biological diversity related to the loss of flat breads. In this context, the submitted manuscript should be considered valuable.
From the manuscript’s content, it seems that the aim was clearly defined. I propose to make some corrections in the Abstrakt, Introduction, and changes in the Material and Methodological part.
(Abstrakt)
Formulated correctly. However, there is a lack of detailed data on the research subject (companies-bakers, associations, experts) and the data collection methods.
(Introduction)
The history of bread is much earlier than any other food. The most common human food on all continents is some form of bread, e.g. made from rye, wheat, beans, potatoes, grass, tree bark, rice, peas, and chestnuts or beechwood. Bread is among the elementary human meals in the oldest sources of the material history of Europe and the Middle East, has been known for more than 10,000 years, and is one of the basic components of the daily human diet. However, it was not bread in the modern sense. Studying the literature, there are many studies on the history of bread production, as mentioned by the authors. The wide-ranging literature in this area shows that it is still unknown who baked the first bread and when. The history of bread is linked to human history. The lack of bread was the cause of violent social movements, its scarcity was the cause of the fall of Rome, the outbreak of the French Revolution as well as the October Revolution in Russia.
However, in describing bread and touching on its cultural and nutritional dimensions (i.e. nutritional and dietary value), the authors ignored its economic importance. The technological and ethnographic aspects are highlighted. Little mention was made of the economic significance, and in this view, bread is defined as a food item made from the milled products of cereals that meet basic human nutritional needs. It is characterised by low income and the price elasticity of demand. It is among the most basic and widespread types of human food. In the marketing classification, it is included in the group of products of frequent purchase, characterised by short shelf life. One of the characteristics of contemporary consumer behaviour and habits is that people often purchase products not because they are used for something but because they symbolise something, as described by Salomon (2006), Sznajder and Goryńska-Goldmann (2012), among others. This principle does not mean that the primary function of a product, in this case, bread, is unimportant, only that the role a product plays in people's lives extends beyond the practical functions it fulfils. Some values are so important that part of the respect due to them flows into their signs. As a result, these signs take on an intrinsic value of their own. These signs are symbols. A symbol, in its visual aspect, becomes a value in itself. It must therefore be treated with the respect due to that which is marked. The symbolic product in many cultures is bread, as the authors of this publication have shown. It is also worth highlighting that its availability provides a sense of security.
I recommend the literature:
- Gül C., Isik H., Bal T., Ozer S. (2003): Bread Consumption and Weste of Households In Urban Area of Adana Province, Electronic Journal of Polish Agricultural Universities, vol.6, issues 2
- Lang T., Barling D., Sharpe R. (2005): Ethical traceability and the UK wheat-flour-bread chain, Agrifood Network, City University London, p. 2 – 5.
- Goryńska-Goldmann E., Sznajder M.: (2012). Selected behaviours and habits of consumers on the bread market. PULS
- Solomon R. Michael (2006): Consumer Behaviour: A European Perspective, Pearson Education, 2006 – 701.
(Materials and Methods)
The part related to data collection needs to be made more specific. Questions to be answered:
- What were the methods of data collection - web survey, telephone survey, face-to-face survey, measurement log, in-depth interview? Identify all tools. Describe the data collection methods used (quantitative and qualitative data?). Was a survey questionnaire used, or was a structured guide also used in the case of in-depth interviews? Please document the advantages and disadvantages of the methods chosen. Indicate that it was necessary to supplement the quantitative data with qualitative research (e.g., aimed at experts) to reflect the nature of the phenomenon studied.
- With whom were the survey questionnaires consulted?
- Were the survey questionnaires pre-tested? How?
- What corrections were made to the final version of the survey questionnaire after pre-testing?
- How many questions did the questionnaire addressed to the bakeries consist of?
- Who was the subject of the survey? The bakery manufacturer? The associations? The expert - who was the expert?
- What criteria were used to select the survey participants?
- Was the enterprise survey completed by the entrepreneurs themselves or with the assistance of a facilitator? What was the situation for the others? What was the average time taken to complete the questionnaires?
- You can add as an annex the questionnaires of the in-depth interview.
Do the Authors have other recommendations for further research that could be prepared based on the presented results? Please additionally explain the limitations of the study.
In general, this article belongs to an interesting literature current. Congratulations on such a rich collection. Please treat the comments and feedback presented more as potential windows for improvement of the paper than as criticism. Good luck.
Author Response
Reviewer 3
Manuskrypt - Foods-1835792-
The Large and Diverse Family of Mediterranean Flat Breads: A Database
This article presents an interesting study on the richness of the bread market from a local perspective, which should be considered in line with the UN's sustainable development agenda. The local food market, with particular reference to the richness of a product group such as bread, represents a wide field of practice and requires work from a practical and scientific perspective. Organisations involved in developing such foods should continually look for ideas to develop wide-ranging cooperation, thinking about how to improve competitiveness within the framework of sustainable development, involving the integration and balancing of various levels, including economic, social, technical, environmental, political and natural. Research is needed to make an 'inventory of products as well as producers'. There should be action to prevent the reduction of cultural and biological diversity related to the loss of flat breads. In this context, the submitted manuscript should be considered valuable.
Answer: We thank the reviewer for appreciating our work and for helpful suggestions. All the points raised have been considered. Modifications have been made in red text.
From the manuscript’s content, it seems that the aim was clearly defined. I propose to make some corrections in the Abstrakt, Introduction, and changes in the Material and Methodological part.
(Abstrakt)
Formulated correctly. However, there is a lack of detailed data on the research subject (companies-bakers, associations, experts) and the data collection methods.
Answer: We added it, although very briefly (see lines 27-28). Please note that we have only added a hint to the subject of the research because max 200 words are allowed in the abstract.
(Introduction)
The history of bread is much earlier than any other food. The most common human food on all continents is some form of bread, e.g. made from rye, wheat, beans, potatoes, grass, tree bark, rice, peas, and chestnuts or beechwood. Bread is among the elementary human meals in the oldest sources of the material history of Europe and the Middle East, has been known for more than 10,000 years, and is one of the basic components of the daily human diet. However, it was not bread in the modern sense. Studying the literature, there are many studies on the history of bread production, as mentioned by the authors. The wide-ranging literature in this area shows that it is still unknown who baked the first bread and when. The history of bread is linked to human history. The lack of bread was the cause of violent social movements, its scarcity was the cause of the fall of Rome, the outbreak of the French Revolution as well as the October Revolution in Russia.
However, in describing bread and touching on its cultural and nutritional dimensions (i.e. nutritional and dietary value), the authors ignored its economic importance. The technological and ethnographic aspects are highlighted. Little mention was made of the economic significance, and in this view, bread is defined as a food item made from the milled products of cereals that meet basic human nutritional needs. It is characterised by low income and the price elasticity of demand.
Answer: Thank you so much for such an interesting view. We totally agree with it, and actually the manuscript already contained a description of the effects of bread/flour shortage, such as the situation in Lebanon, Egypt and Jordan (see lines 282-284 and 318-324, in green). The economic importance of bread has been briefly highlighted (see lines 316-318), by incorporating the suggestions of the Reviewer into other economic considerations already made in the original draft (green text). Actually, a thorough treatment of the economic importance of bread was beyond the scope of the paper and, above all, the expertise of the authors. We would be happy to collaborate!
It is among the most basic and widespread types of human food. In the marketing classification, it is included in the group of products of frequent purchase, characterised by short shelf life. One of the characteristics of contemporary consumer behaviour and habits is that people often purchase products not because they are used for something but because they symbolise something, as described by Salomon (2006), Sznajder and Goryńska-Goldmann (2012), among others. This principle does not mean that the primary function of a product, in this case, bread, is unimportant, only that the role a product plays in people's lives extends beyond the practical functions it fulfils. Some values are so important that part of the respect due to them flows into their signs. As a result, these signs take on an intrinsic value of their own. These signs are symbols. A symbol, in its visual aspect, becomes a value in itself. It must therefore be treated with the respect due to that which is marked. The symbolic product in many cultures is bread, as the authors of this publication have shown. It is also worth highlighting that its availability provides a sense of security.
Answer: We sincerely thank a lot the Reviewer for these considerations, which have been incorporated into the discussion (see lines 587-596). We would be pleased to collaborate with him/her on this topic maybe in future research.
I recommend the literature:
- Gül C., Isik H., Bal T., Ozer S. (2003): Bread Consumption and Waste of Households In Urban Area of Adana Province, Electronic Journal of Polish Agricultural Universities, vol.6, issues 2
Answer: Added, ref. 68.
- Lang T., Barling D., Sharpe R. (2005): Ethical traceability and the UK wheat-flour-bread chain, Agrifood Network, City University London, p. 2 – 5.
Answer: Added, ref. 18
- Goryńska-Goldmann E., Sznajder M.: (2012). Selected behaviours and habits of consumers on the bread market. PULS
Answer: This specific reference was not retrievable. It has been added another one, similar (ref. mo. 15).
- Solomon R. Michael (2006): Consumer Behaviour: A European Perspective, Pearson Education, 2006 – 701. Added (ref. 73).
(Materials and Methods)
The part related to data collection needs to be made more specific. Questions to be answered:
- What were the methods of data collection - web survey, telephone survey, face-to-face survey, measurement log, in-depth interview? Identify all tools. Describe the data collection methods used (quantitative and qualitative data?). Was a survey questionnaire used, or was a structured guide also used in the case of in-depth interviews? Please document the advantages and disadvantages of the methods chosen. Indicate that it was necessary to supplement the quantitative data with qualitative research (e.g., aimed at experts) to reflect the nature of the phenomenon studied.
Answer: Sorry we described the methods of data collection too briefly and with few details. We implemented a detailed description of the data collection procedure (see lines 126-159).
- With whom were the survey questionnaires consulted?
Answer: The questionnaire was prepared according to previous research of the authors and current literature and was consulted with the President of a consortium of bakers specialized in the production of Focaccia barese flat bread (see line 145).
- Were the survey questionnaires pre-tested? How?
Answer: The questionnaire was pre-tested with the President of a consortium of bakers specialized in the production of Focaccia barese flat bread, who was asked to answer the questions and comment on their feasibility (too generic, or too specific, or too technical, difficult to understand). We added this detail to the text (see line 145).
- What corrections were made to the final version of the survey questionnaire after pre-testing?
Answer: We deleted specific questions on packaging material and modified atmosphere or storage conditions, too far from the common knowledge of the bakers. We added this detail to the text (see lines 148-149).
- How many questions did the questionnaire addressed to the bakeries consist of?
Answer: The questionnaire was composed of 20 questions.
- Who was the subject of the survey? The bakery manufacturer? The associations? The expert - who was the expert?
Answer: According to the local availability, the subject was primarily bakers’ associations.
- What criteria were used to select the survey participants?
Answer: Convenience choice was used for the selection of study participants.
- Was the enterprise survey completed by the entrepreneurs themselves or with the assistance of a facilitator? What was the situation for the others? What was the average time taken to complete the questionnaires?
Answer: The enterprise was always assisted by facilitators, who were the researchers involved in the study. The average time to complete the questionnaire was approximatively 30 minutes.
- You can add as an annex the questionnaires of the in-depth interview
Answer: Yes, we added the questionnaire as supplementary material.
- Do the Authors have other recommendations for further research that could be prepared based on the presented results?
Answer: Yes, to extend the database to other countries. “Regarding the periodical survey, it is worth noting that the database could be updated as well extended to include many other countries where flat breads are produced”. Lines 760-761.
- Please additionally explain the limitations of the study.
Answer: The limitations of the study were already explained in the original draft of the article. Please see green lines at 314-328.
In general, this article belongs to an interesting literature current. Congratulations on such a rich collection. Please treat the comments and feedback presented more as potential windows for improvement of the paper than as criticism. Good luck.
Thank you so much

Reviewer 4 Report
In this paper, based on publicly available databases, an overview of the main characteristics and distribution of the original flatbreads in the nine Mediterranean countries is presented. However, the number of samples for some groups (referring to different countries or other categories) is insufficient compared to others that are excessive in this situation, but if the random selection of the presented samples follows the distribution of these products in the market, this should be explained.
It is expected that the results obtained will contribute to the nutritional and technical improvement of flatbreads, but this part is very general. The promotion of traditional products is welcomed, but again, few countries have been studied on this topic.
Figure 10 should be improved.
Examples of the main types of flatbreads from all the countries studied are not evenly represented.
Author Response
Reviewer 4
In this paper, based on publicly available databases, an overview of the main characteristics and distribution of the original flatbreads in the nine Mediterranean countries is presented. However, the number of samples for some groups (referring to different countries or other categories) is insufficient compared to others that are excessive in this situation, but if the random selection of the presented samples follows the distribution of these products in the market, this should be explained.
Answer: Sorry for being unclear. There was no random selection, the survey included all the flat breads of the examined countries, no one excluded, and the strongest point of this work is exactly the accuracy in the collection of data, made with the help of experts in each country, to be sure of not excluding any product. We modified the sentence to make it clearer (see lines 126-127).
It is expected that the results obtained will contribute to the nutritional and technical improvement of flatbreads, but this part is very general.
Answer: The results obtained will contribute to the selection of the most suitable flat breads for nutritional and technical improvement, but not within this research. This is expected in future research, according to the criteria described at lines 743-750.
The promotion of traditional products is welcomed, but again, few countries have been studied on this topic.
Answer: We agree, we know that this is a limitation of this study, which is derived from a funded project that funded only these nine countries. Hopefully the study will be extended in the future. The paper has established a working method which could easily applied to other countries to extend the database. We added a sentence to highlight that future extension of the study, to increase the number of countries involved, is advisable (see lines 760-761).
Figure 10 should be improved.
Answer: Fig. 10 was improved.
Examples of the main types of flatbreads from all the countries studied are not evenly represented.
Answer: Sorry we were unclear. The survey included all the flat breads of the examined countries, no one excluded (see lines 126-127), so the different number reflected the real situation, we cannot change it.
